# BESPOKE SOLVERS FOR GENERATIVE FLOW MODELS

**N. Shaul**[1]  **J. Perez**[2]  **R. T. Q. Chen**[3]  **A. Thabet**[2]  **A. Pumarola**[2]  **Y. Lipman**[3,1]
[1]Weizmann Institute of Science   [2]GenAI, Meta   [3]FAIR, Meta

## ABSTRACT

Diffusion or flow-based models are powerful generative paradigms that are notoriously hard to sample as samples are defined as solutions to high-dimensional Ordinary or Stochastic Differential Equations (ODEs/SDEs) which require a large Number of Function Evaluations (NFE) to approximate well. Existing methods to alleviate the costly sampling process include model distillation and designing dedicated ODE solvers. However, distillation is costly to train and sometimes can deteriorate quality, while dedicated solvers still require relatively large NFE to produce high quality samples. In this paper we introduce *"Bespoke solvers"*, a novel framework for constructing custom ODE solvers tailored to the ODE of a given pre-trained flow model. Our approach optimizes an order consistent and parameter-efficient solver (*e.g.*, with 80 learnable parameters), is trained for roughly 1% of the GPU time required for training the pre-trained model, and significantly improves approximation and generation quality compared to dedicated solvers. For example, a Bespoke solver for a CIFAR10 model produces samples with Fréchet Inception Distance (FID) of 2.73 with 10 NFE, and gets to 1% of the Ground Truth (GT) FID (2.59) for this model with only 20 NFE. On the more challenging ImageNet-64×64, Bespoke samples at 2.2 FID with 10 NFE, and gets within 2% of GT FID (1.71) with 20 NFE.

## 1 INTRODUCTION

Diffusion models (Sohl-Dickstein et al., 2015; Ho et al., 2020), and more generally flow-based models (Song et al., 2020b; Lipman et al., 2022; Albergo & Vanden-Eijnden, 2022), have become prominent in generation of images (Dhariwal & Nichol, 2021; Rombach et al., 2021), audio (Kong et al., 2020; Le et al., 2023), and molecules (Kong et al., 2020). While training flow models is relatively scalable and efficient, sampling from a flow-based model entails solving a Stochastic or Ordinary Differential Equation (SDE/ODE) in high dimensions, tracing a velocity field defined with the trained neural network. Using off-the-shelf solvers to approximate the solution of this ODE to a high precision requires a large Number (*i.e.*, 100s) of Function Evaluations (NFE), making sampling one of the main standing challenges in flow models. Improving the sampling complexity of flow models, without degrading sample quality, will open up new applications that require fast sampling, and will help reducing the carbon footprint and deployment cost of these models.

Current approaches for efficient sampling of flow models divide into two main groups: (i) *Distillation*: where the pre-trained model is fine-tuned to predict either the final sampling (Luhman & Luhman, 2021) or some intermediate solution steps (Salimans & Ho, 2022) of the ODE. Distillation does not guarantee sampling from the pre-trained model's distribution, but, when given access to the training data during distillation training, it is shown to empirically generate samples of comparable quality to the original model (Salimans & Ho, 2022; Meng et al., 2023). Unfortunately, the GPU time required to distill a model is comparable to the training time of the original model Salimans & Ho (2022), which is often considerable. (ii) *Dedicated solvers*: where the specific structure of the ODE is used to design a more efficient solver (Song et al., 2020a; Lu et al., 2022a;b) and/or employ a suitable solver family from the literature of numerical analysis (Zhang & Chen, 2022; Zhang et al., 2023). The main benefit of this approach is two-fold: First, it is *consistent*, *i.e.*, as the number of steps (NFE) increases, the samples converge to those of the pre-trained model. Second, it does not require further training/fine-tuning of the pre-trained model, consequently avoiding long additional training times and access to training data. Related to our approach, some works have tried to learn an ODE solver within a certain class (Watson et al., 2021; Duan et al., 2023); however, they do not guarantee consistency and usually introduce moderate improvements over generic dedicated solvers.

In this paper, we introduce *Bespoke solvers*, a framework for learning consistent ODE solvers *custom-tailored* to pre-trained flow models. The main motivation for Bespoke solvers is that different models exhibit sampling paths with different characteristics, leading to local truncation errors that are specific to each instance of a trained model. A key observation of this paper is that optimizing a solver for a particular model can significantly improve quality of samples for low NFE compared to existing dedicated solvers. Furthermore, Bespoke solvers use a very small number of learnable parameters and consequently are efficient to train. For example, we have trained $n \in \{5, 8, 10\}$ steps Bespoke solvers for a pre-trained ImageNet-64×64 flow model with $\{40, 64, 80\}$ learnable parameters (resp.) producing images with Fréchet Inception Distances (FID) of 2.2, 1.79, 1.71 (resp.), where the latter is within 2% from the Ground Truth (GT) FID (1.68) computed with $\sim 180$ NFE. The Bespoke solvers were trained (using a rather naive implementation) for roughly 1% of the GPU time required for training the original model. Figure 1 compares sampling at 10 NFE from a pre-trained AFHQ-256×256 flow model with order 2 Runge-Kutta (RK2) and its Bespoke version (RK2-Bes), along with the GT sample that requires $\sim 180$ NFE. Our work brings the following contributions:

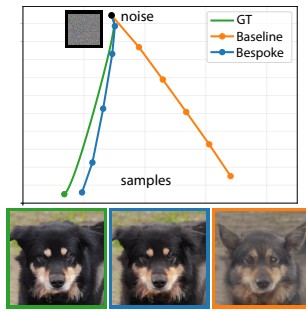

**Figure 1:** Using 10 NFE to sample using our Bespoke solver improves fidelity w.r.t. the baseline (RK2) solver. Visualization of paths was done with the 2D PCA plane approximating the noise and end sample points.

1. A differentiable parametric family of consistent ODE solvers.

2. A tractable loss that bounds the global truncation error while allowing parallel computation.

3. An algorithm for training a Bespoke $n$-step solver for a specific pre-trained model.

4. Significant improvement over dedicated solvers in generation quality for low NFE.

## 2 BESPOKE SOLVERS

We consider a pre-trained flow model taking some prior distribution (noise) $p$ to a target (data) distribution $q$ in data space $\mathbb{R}^d$. The flow model (Chen et al., 2018) is represented by a time-dependent Vector Field (VF) $u : [0, 1] \times \mathbb{R}^d \to \mathbb{R}^d$ that transforms a noise sample $x_0 \sim p(x_0)$ to a data sample $x_1 \sim q(x_1)$ by solving the ODE

$$\dot{x}(t) = u_t(x(t)), \tag{1}$$

with the initial condition $x(0) = x_0 \sim p(x_0)$, from time $t = 0$ until time $t = 1$, and $\dot{x}(t) := \frac{d}{dt}x(t)$. The solution at time $t = 1$, *i.e.*, $x(1) \sim q(x(1))$, is the generated target sample.

**Numerical ODE solvers.** Solving equation 1 is done in practice with numerical ODE solvers. A numerical solver is defined by an update rule:

$$(t_{\text{next}}, x_{\text{next}}) = \text{step}(t, x; u_t). \tag{2}$$

---

**Algorithm 1** Numerical ODE solver.

**Require:** $t_0, x_0$
    **for** $i = 0, 1, \ldots, n-1$ **do**
        $(t_{i+1}, x_{i+1}) = \text{step}(t_i, x_i; u_t)$
    **end for**
    **return** $x_n$

---

The update rule takes as input current time $t$ and approximate solution $x$, and outputs the next time step $t_{\text{next}}$ and the corresponding approximation $x_{\text{next}}$ to the true solution $x(t_{\text{next}})$ at time $t_{\text{next}}$. To approximate the solution at some desired end time, *i.e.*, $t = 1$, one first initializes the solution at $t = 0$ and repeatedly applies the update rule in equation 2 $n$ times, as presented in Algorithm 1. The step is designed so that $t_n = 1$.

An ODE solver (step) is said to be of *order* $k$ if its local truncation error is

$$\|x(t_{\text{next}}) - x_{\text{next}}\| = O\left((t_{\text{next}} - t)^{k+1}\right), \tag{3}$$

asymptotically as $t_{\text{next}} \to t$, where $t \in [0, 1)$ is arbitrary but fixed and $t_{\text{next}}, x_{\text{next}}$ are defined by the solver, equation 2. A popular family of solvers that offers a wide range of orders is the Runge-Kutta (RK) family (Iserles, 2009). Two of the most popular members of the RK family are (set $h = n^{-1}$):

RK1 (Euler - order 1): $\quad \text{step}(t, x; u_t) = (t + h, \, x + h u_t(x)), \tag{4}$

RK2 (Midpoint - order 2): $\quad \text{step}(t, x; u_t) = \left(t + h, \, x + h u_{t + \frac{h}{2}}\left(x + \frac{h}{2}u_t(x)\right)\right). \tag{5}$

**Approach outline.** Given a pre-trained $u_t$ and a target number of time steps $n$ our goal is to find a custom (Bespoke) solver that is optimal for approximating the samples $x(1)$ defined via equation 1 from initial conditions sampled according to $x(0) = x_0 \sim p(x_0)$. To that end we develop two components: (i) a differentiable parametric family of update rules $\text{step}^\theta$, with parameters $\theta \in \mathbb{R}^p$

---
**Algorithm 2** Bespoke solver.
---
**Require:** $t_0, x_0$, pre-trained $u_t, \theta$
  **for** $i = 0, 1, \ldots, n - 1$ **do**
    $(t_{i+1}, x_{i+1}) \leftarrow \text{step}^\theta(t_i, x_i; u_t)$
  **end for**
  **return** $x_n$
---

(where $p$ is *very* small), where sampling is done by replacing step with $\text{step}^\theta$ in Algorithm 1, see Algorithm 2; and (ii) a tractable loss bounding the *global truncation error*, i.e., the Root Mean Square Error (RMSE) between the approximate sample $x_n^\theta$ and the GT sample $x(1)$,

$$\text{Global truncation error:} \quad \mathcal{L}_{\text{RMSE}}(\theta) = \mathbb{E}_{x_0 \sim p(x_0)} \left\| x(1) - x_n^\theta \right\|, \tag{6}$$

where $x_n^\theta$ is the output of Algorithm 2, and $\|x\| = (\frac{1}{d} \sum_{j=1}^d [x^{(j)}]^2)^{1/2}$.

## 2.1 PARAMETRIC FAMILY OF ODE SOLVERS THROUGH TRANSFORMED SAMPLING PATHS

Our strategy for defining the parametric family of solvers $\text{step}^\theta$ is using a generic base ODE solver, such as RK1 or RK2, applied to a parametric family of *transformed paths*.

**Transformed sampling paths.** We transform the sample trajectories $x(t)$ by applying two components: a time reparametrization and an arbitrary invertible transformation. That is,

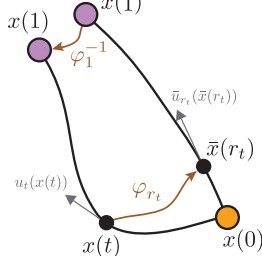

$$\bar{x}(r) = \varphi_r(x(t_r)), \quad r \in [0, 1], \tag{7}$$

where $t_r, \varphi_r(x)$ are arbitrary functions in a family $\mathcal{F}$ defined by the following conditions: (i) *Smoothness*: $t_r : [0, 1] \to [0, 1]$ is a diffeomorphism[1], and $\varphi : [0, 1] \times \mathbb{R}^d \to \mathbb{R}^d$ is $C^1$ and a diffeomorphism in $x$. We also assume $r_t$ and $\varphi_r^{-1}$ are Lipschitz continuous with a constant $L > 0$. (ii) *Boundary conditions*: $t_r$ satisfies $t_0 = 0$ and $t_1 = 1$, and $\varphi_0(\cdot)$ is the identity function, i.e., $\varphi_0(x) = x$ for all $x \in \mathbb{R}^d$. Fig-

**Figure 2:** Transformed paths.

ure 2 depicts a transformation of a path, $x(t)$. Note that $\bar{x}(0) = x(0)$, however the end point $\bar{x}(1)$ does not have to coincide with $x(1)$. Furthermore, as $t_r : [0, 1] \to [0, 1]$ is a diffeomorphism, $t_r$ is strictly monotonically increasing.

The motivation behind the definition of the transformed trajectories is that it allows reconstructing $x(t)$ from $\bar{x}(r)$. Indeed, denoting $r = r_t$ the inverse function of $t = t_r$ we have

$$x(t) = \varphi_{r_t}^{-1}(\bar{x}(r_t)). \tag{8}$$

Our hope is to find a transformation that simplifies sampling paths and allows the base solver to provide better approximations of the GT samples. The transformed trajectory $\bar{x}_r$ is defined by a VF $\bar{u}_r(x)$ that can be given an explicit form as follows (proof in Appendix A):

**Proposition 2.1.** *Let $x(t)$ be a solution to equation 1. Denote $\dot{\varphi}_r := \frac{d}{dr}\varphi_r$ and $\dot{t}_r := \frac{d}{dr}t_r$. Then $\bar{x}(r)$ defined in equation 7 is a solution to the ODE (equation 1) with the VF*

$$\bar{u}_r(x) = \dot{\varphi}_r(\varphi_r^{-1}(x)) + \dot{t}_r \partial_x \varphi_r(\varphi_r^{-1}(x)) u_{t_r}(\varphi_r^{-1}(x)). \tag{9}$$

**Solvers via transformed paths.** We are now ready to define our parametric family of solvers $\text{step}^\theta(t, x; u_t)$: First we transform the input sample $(t, x)$ according to equation 7 to

$$(r, \bar{x}) = (r_t, \varphi_{r_t}(x)). \tag{10}$$

Next, we perform a step with the base solver of choice,

$$(r_{\text{next}}, \bar{x}_{\text{next}}) = \text{step}(r, \bar{x}; \bar{u}_r), \tag{11}$$

and lastly, transform back using equation 8 to define the parametric solver $\text{step}^\theta$ via

$$(t_{\text{next}}, x_{\text{next}}) = \text{step}^\theta(x, t; u_t) = \left( t_{r_{\text{next}}}, \varphi_{r_{\text{next}}}^{-1}(\bar{x}_{\text{next}}) \right). \tag{12}$$

The parameters $\theta$ denote the parameterized transformations $t_r$ and $\varphi_r$ satisfying the properties of $\mathcal{F}$ and the choice of a base solver step. In Section 2.2 we derive the explicit rules we use in this paper.

---

[1] A diffeomorphism is a $C^1$ continuously differentiable function with a $C^1$ continuous differentiable inverse.

**Consistency of solvers.** An important property of the parametric solver $\text{step}^\theta$ is *consistency*. Namely, due to the properties of $\mathcal{F}$, regardless of the particular choice of $t_r, \varphi_r \in \mathcal{F}$, the solver $\text{step}^\theta$ has the same local truncation error as the base solver.

**Theorem 2.2.** *(Consistency of parametric solvers) Given arbitrary $t_r, \varphi_r$ in the family of functions $\mathcal{F}$ and a base ODE solver of order $k$, the corresponding ODE solver $\text{step}^\theta$ is also of order $k$, i.e.,*

$$\|x(t_{next}) - x_{next}\| = O((t_{next} - t)^{k+1}). \tag{13}$$

The proof is provided in Appendix B. Therefore, as long as $t_r, \varphi_r(x)$ are in $\mathcal{F}$, decreasing the base solver's step size $h \to 0$ will result in our approximated sample $x_n^\theta$ converging to the exact sample $x(1)$ of the trained model in the limit, *i.e.*, $x_n^\theta \to x(1)$ as $n \to \infty$.

## 2.2 TWO USE CASES

We instantiate the Bespoke solver framework for two cases of interest (a full derivation is in Appendix E), and later prove that our choice of transformations in fact covers all "noise scheduler" configurations used in the standard diffusion model literature. In our use cases, we consider a time-dependent scaling as our invertible transformation $\varphi_r$,

$$\varphi_r(x) = s_r x, \text{ and its inverse } \varphi_r^{-1}(x) = x/s_r, \tag{14}$$

where $s : [0, 1] \to \mathbb{R}_{>0}$ is a strictly positive $C^1$ scaling function such that $s_0 = 1$ (*i.e.*, satisfying the boundary condition of $\varphi$). The transformation of trajectories, *i.e.*, equations 7 and 8, take the form

$$\bar{x}(r) = s_r x(t_r), \text{ and } x(t) = \bar{x}(r_t)/s_{t_r}, \tag{15}$$

and we name this transformation: *scale-time*. The transformed VF $\bar{u}_r$ (equation 9) is thus

$$\bar{u}_r(x) = \frac{\dot{s}_r}{s_r} x + \dot{t}_r s_r u_{t_r}\left(\frac{x}{s_r}\right). \tag{16}$$

**Use case I: RK1-Bespoke.** We consider RK1 (Euler) method (equation 4) as the base solver step and denote $r_i = ih$, $i \in [n]$, where $[n] = \{0, 1, \ldots, n\}$ and $h = n^{-1}$. Substituting equation 4 in equation 11, we get from equation 12 that

$$\text{step}^\theta(t_i, x_i; u_t) := \left(t_{i+1}, \frac{s_i + h\dot{s}_i}{s_{i+1}} x_i + h\dot{t}_i \frac{s_i}{s_{i+1}} u_{t_i}(x_i)\right), \tag{17}$$

where we denote $t_i = t_{r_i}$, $\dot{t}_i = \frac{d}{dr}|_{r=r_i} t_r$, $s_i = s_{r_i}$, $\dot{s}_i = \frac{d}{dr}|_{r=r_i} s_r$, and $i \in [n-1]$. The learnable parameters $\theta \in \mathbb{R}^p$ and their constraints are derived from the fact that the functions $t_r, \varphi_r$ are members of $\mathcal{F}$. There are $p = 4n - 1$ parameters in total: $\theta = (\theta^t, \theta^s)$, where

$$\theta^t : \begin{cases} 0 = t_0 < t_1 < \cdots < t_{n-1} < t_n = 1 \\ \dot{t}_0, \ldots, \dot{t}_{n-1} > 0 \end{cases} \quad, \quad \theta^s : \begin{cases} s_1, \ldots, s_n > 0 \;, \; s_0 = 1 \\ \dot{s}_0, \ldots, \dot{s}_{n-1} \end{cases} \quad . \tag{18}$$

Note that we ignore the Lipschitz constant constraints in $\mathcal{F}$ when deriving the constraints for $\theta$.

**Use case II: RK2-Bespoke.** Here we choose the RK2 (Midpoint) method (equation 5) as the base solver step. Similarly to the above, substituting equation 5 in equation 11, we get

$$\text{step}^\theta(t_i, x_i; u_t) := \left(t_{i+1}, \frac{s_i}{s_{i+1}} x_i + \frac{h}{s_{i+1}} \left\{ \frac{\dot{s}_{i+\frac{1}{2}}}{s_{i+\frac{1}{2}}} z_i + \dot{t}_{i+\frac{1}{2}} s_{i+\frac{1}{2}} u_{t_{i+\frac{1}{2}}}\left(\frac{z_i}{s_{i+\frac{1}{2}}}\right) \right\}\right), \tag{19}$$

where we set $r_{i+\frac{1}{2}} = r_i + \frac{h}{2}$, and accordingly $t_{i+\frac{1}{2}}, \dot{t}_{i+\frac{1}{2}}, s_{i+\frac{1}{2}}$, and $\dot{s}_{i+\frac{1}{2}}$ are defined, and

$$z_i = \left(s_i + \frac{h}{2}\dot{s}_i\right) x_i + \frac{h}{2} s_i \dot{t}_i u_{t_i}(x_i). \tag{20}$$

In this case there are $p = 8n - 1$ learnable parameters, $\theta = (\theta^t, \theta^s) \in \mathbb{R}^p$, where

$$\theta^t : \begin{cases} 0 = t_0 < t_{\frac{1}{2}} < \cdots < t_n = 1 \\ \dot{t}_0, \dot{t}_{\frac{1}{2}}, \ldots, \dot{t}_{n-1}, \dot{t}_{n-\frac{1}{2}} > 0 \end{cases} \quad, \quad \theta^s : \begin{cases} s_{\frac{1}{2}}, s_1, \ldots, s_n > 0 \;, \; s_0 = 1 \\ \dot{s}_0, \dot{s}_{\frac{1}{2}}, \ldots, \dot{s}_{n-\frac{1}{2}} \end{cases} \quad . \tag{21}$$

**Equivalence of scale-time transformations and Gaussian Paths.** We note that our *scale-time* transformation covers *all* possible trajectories used by diffusion and flow models trained with Gaussian distributions. Denote by $p_t(x)$ the probability density function of the random variable $x(t)$, where $x(t)$ is defined by a random initial sampling $x(0) = x_0 \sim p(x_0)$ and solving the ODE in equation 1.

When training a Diffusion or Flow Matching models, $p_t$ has the form $p_t(x) = \int p_t(x|x_1)q(x_1)dx_1$, where $p_t(x|x_1) = \mathcal{N}(x|\alpha_t x_1, \sigma_t^2 I)$. A pair of functions $\alpha, \sigma : [0,1] \to [0,1]$ satisfying

$$\alpha_0 = 0 = \sigma_1, \quad \alpha_1 = 1 = \sigma_0, \quad \text{and strictly monotonic } \mathrm{snr}(t) = \alpha_t/\sigma_t \tag{22}$$

is called a *scheduler*[2]. We use the term *Gaussian Paths* for the collection of probability paths $p_t(x)$ achieved by different schedulers. The velocity vector field that generates $p_t(x)$ and results from zero Diffusion/Flow Matching training loss is

$$u_t(x) = \int u_t(x|x_1) \frac{p_t(x|x_1)q(x_1)}{p_t(x)} dx_1, \tag{23}$$

where $u_t(x|x_1) = \frac{\dot{\sigma}_t}{\sigma_t}x + \left[\dot{\alpha}_t - \dot{\sigma}_t \frac{\alpha_t}{\sigma_t}\right]x_1$, as derived in Lipman et al. (2022). Next, we generalize a result by Kingma et al. (2021) and Karras et al. (2022) to consider marginal sampling paths $x(t)$ defined by $u_t(x)$, and show that any two such paths are related by a scale-time transformation:

**Theorem 2.3.** *(Equivalence of Gaussian Paths and scale-time transformation) Consider a Gaussian Path defined by a scheduler $(\alpha_t, \sigma_t)$, and let $x(t)$ denote the solution of equation 1 with $u_t$ defined in equation 23 and initial condition $x(0) = x_0$. Then,*

*(i) For every other Gaussian Path defined by a scheduler $(\bar{\alpha}_r, \bar{\sigma}_r)$ with trajectories $\bar{x}(r)$ there exists a scale-time transformation with $s_1 = 1$ such that $\bar{x}(r) = s_r x(t_r)$.*

*(ii) For every scale-time transformation with $s_1 = 1$ there exists a Gaussian Path defined by a scheduler $(\bar{\alpha}_r, \bar{\sigma}_r)$ with trajectories $\bar{x}(r)$ such that $s_r x(t_r) = \bar{x}(r)$.*

(Proof in Appendix C.) Assuming an ideal velocity field (equation 23), *i.e.*, the pre-trained model is optimal, this theorem implies that searching over the scale-time transformations is equivalent to searching over all possible Gaussian Paths. Note, that in practice we allow $s_1 \neq 1$, expanding beyond the standard space of Gaussian Paths. Another interesting consequence of Theorem 2.3 (simply plug in $t = 1$) is that all ideal velocity fields in equation 23 define the *same* coupling, *i.e.*, joint distribution, of noise $x_0$ and data $x_1$.

## 2.3 RMSE UPPER BOUND LOSS

Optimizing directly the RMSE loss (equation 6) is theoretically possible but would require keeping a full computational graph of Algorithm 2, *i.e.*, $n\times$order compositions of $u_t$ leading to a large memory footprint. Therefore, we instead derive an *upper-bound* to the RMSE loss that enables *parallel computation* over the steps of the solver, considerably reducing memory consumption. To construct the bound, let us fix an initial condition $x_0 \sim p(x_0)$ and denote as before $x(1)$ to be the exact solution of the sample path (equation 1). Furthermore, consider a candidate solver $\mathrm{step}^\theta$, and denote its $t$ and $x$ coordinate updates by $\mathrm{step}^\theta = (\mathrm{step}_t^\theta, \mathrm{step}_x^\theta)$. Applying Algorithm 2 with $t_0 = 0, x_0$ produces a series of approximations $x_i^\theta$, each corresponds to a time step $t_i$, $i \in [n]$. Lastly, we denote by

$$e_i^\theta = \left\|x(t_i) - x_i^\theta\right\|, \quad d_i^\theta = \left\|x(t_i) - \mathrm{step}_x^\theta(t_{i-1}, x(t_{i-1}); u_t)\right\| \tag{24}$$

the *global* and *local* truncation errors at time $t_i$, respectively. Our goal is to bound the global error at the final time step $t_n = 1$, *i.e.*, $e_n^\theta$. Using the update rule definition (equation 2) and triangle inequality we can bound

$$e_{i+1}^\theta \leq \left\|x(t_{i+1}) - \mathrm{step}_x^\theta(t_i, x(t_i); u_t)\right\| + \left\|\mathrm{step}_x^\theta(t_i, x(t_i); u_t) - \mathrm{step}_x^\theta(t_i, x_i^\theta; u_t)\right\| \leq d_{i+1}^\theta + L_i^\theta e_i^\theta,$$

where $L_i^\theta$ is defined to be the Lipschitz constant of the function $\mathrm{step}_x^\theta(t_i, \cdot\,; u_t)$. To simplify notation we set by definition $L_n^\theta = 1$ (this is possible since $L_n^\theta$ does not actually participate in the bound).

---

[2]We use the convention of noise at time $t = 0$ and data at time $t = 1$.

Using the above bound $n$ times and noting that $e_0^\theta = 0$ we get

$$e_n^\theta \leq \sum_{i=1}^{n} M_i^\theta d_i^\theta, \text{ where } M_i^\theta = \prod_{j=i}^{n} L_j^\theta. \quad (25)$$

Motivated by this bound we define our RMSE-Bound loss:

$$\mathcal{L}_{\text{RMSE-B}}(\theta) = \mathbb{E}_{x_0 \sim p(x_0)} \sum_{i=1}^{n} M_i^\theta d_i^\theta, \quad (26)$$

---

**Algorithm 3** Bespoke training.

**Require:** pre-trained $u_t$, number of steps $n$
  initialize $\theta \in \mathbb{R}^p$
  **while** not converged **do**
    $x_0 \sim p(x_0)$         ▷ sample noise
    $x(t) \leftarrow$ solve ODE 1     ▷ GT path
    $\mathcal{L} \leftarrow 0$            ▷ init loss
    **parallel for** $i = 0, ..., n - 1$ **do**
      $x_{i+1}^\theta \leftarrow \text{step}_x^\theta(x_i^{\text{aux}}(t_i), t_i; u_t)$
      $\mathcal{L} += M_{i+1}^\theta \| x_{i+1}^{\text{aux}}(t_{i+1}) - x_{i+1}^\theta \|$
    **end for**
    $\theta \leftarrow \theta - \gamma \nabla_\theta \mathcal{L}$   ▷ optimization step
  **end while**
  **return** $\theta$

---

where $d_i^\theta$ is defined in equation 24 and $M_i^\theta$ defined in equation 25. The constants $L_i^\theta$ depend both on the parameters $\theta$ and the Lipschitz constant $L_u$ of the network $u_t$. As $L_u$ is difficult to estimate, we treat $L_u$ as a hyper-parameter, denoted $L_\tau$ (in all experiments we use $L_\tau = 1$), and compute $L_i^\theta$ in terms of $\theta$ and $L_\tau$ for our two Bespoke solvers, RK1 and RK2, in Appendix D. Assuming that $L_\tau \geq L_u$, an immediate consequence of the bound in equation 25 is that the RMSE-Bound loss bounds the RMSE loss, *i.e.*, the global truncation error defined in equation 6, $\mathcal{L}_{\text{RMSE}}(\theta) \leq \mathcal{L}_{\text{RMSE-B}}(\theta)$.

**Implementation of the RMSE-Bound loss.** We provide pseudocode for Bespoke training in Algorithm 3. During training, we need to have access to the GT path $x(t)$ at times $t_i$, $i \in [n]$, which we compute with a generic solver. The Bespoke loss is constructed by plugging step$^\theta$ (equations 17 or 19) into $d_i$ (equation 24). The gradient $\nabla_\theta \mathcal{L}_{\text{RMSE-B}}(\theta)$ requires the derivatives $\partial x(t_i)/\partial t_i$. Computing the derivatives of $x(t_i)$ can be done using the ODE it obeys, *i.e.*, $\dot{x}(t_i) = u_{t_i}(x_i)$. Therefore, a simple way to write the loss ensuring correct gradients w.r.t. $t_i$ is replace $x(t_i)$ with $x_i^{\text{aux}}(t_i)$ where

$$x_i^{\text{aux}}(t) = x(\llbracket t_i \rrbracket) + u_{\llbracket t_i \rrbracket}(x(\llbracket t_i \rrbracket))(t - \llbracket t_i \rrbracket), \quad (27)$$

where $\llbracket \cdot \rrbracket$ denotes the stop gradient operator; *i.e.*, $x_i^{\text{aux}}(t)$ is linear in $t$ and its value and derivative w.r.t. $t$ coincide with that of $x(t_i)$ at time $t = t_i$. Full details are provided in Appendix F. In Appendix K.1 we provide an ablation experiment comparing different Bespoke losses and corresponding algorithms including the direct RMSE loss (eq. 6) and our RMSE-Bound loss (eq. 26).

## 3 PREVIOUS WORK

Diffusion models (Sohl-Dickstein et al., 2015; Ho et al., 2020) are a powerful paradigm for generative models that for sampling require solving a Stochastic Differential Equation (SDE), or its associated ODE, describing a (deterministic) flow process (Song et al., 2020a). Diffusion models have been generalized to paradigms directly aiming to learn a deterministic flow (Lipman et al., 2022; Albergo & Vanden-Eijnden, 2022; Liu et al., 2022). Flow-based models are efficient to train but costly to sample. Previous works had tackled the sample complexity of flow models by building *dedicated solver schemes* and *distillation*.

**Dedicated Solvers.** This line of works introduced specialized ODE solvers exploiting the structure of the sampling ODE. Lu et al. (2022a); Zhang & Chen (2022) utilize the semi-linear structure of the score/$\epsilon$-based sampling ODE to adopt a method of exponential integrators. (Zhang et al., 2023) further introduced refined error conditions to fulfill desired order conditions and achieve better sampling, while Lu et al. (2022b) adapted the method to guided sampling. Karras et al. (2022) suggested transforming the ODE to sample a different Gaussian Path for more efficient sampling, while also suggesting non-uniform time steps. In principle, all of these methods effectively proposed—based on intuition and heuristics—to apply a particular scale-time transformation to the sampling trajectories of the pre-trained model for more efficient sampling, while Bespoke solvers search for an *optimal* transformation within the entire space of scale-time transformations.

Other works also aimed at learning the solver: Dockhorn et al. (2022) (GENIE) introduced a higher-order solver, and distilled the necessary JVP for their method; Watson et al. (2021) (DDSS) optimized a perceptual loss considering a family of generalized Gaussian diffusion models; Lam et al. (2021) improved the denoising process using bilateral filters, thereby indirectly affecting the efficiency of the ODE solver; Duan et al. (2023) suggested to learn a solver for diffusion models by replacing every other function evaluation by a linear subspace projection. Our Bespoke Solvers belong to this family of learnt solvers, however, they are consistent by construction (Theorem 2.2) and minimize a bound on the solution error (for the appropriate Lipschitz constant parameter).

**Distillation.** Distillation techniques aim to simplify sampling from a trained model by fine-tuning or training a new model to produce samples with fewer function evaluations. Luhman & Luhman (2021) directly regressed the trained model's samples, while Salimans & Ho (2022); Meng et al. (2023) built a sequence of models each reducing the sampling complexity by a factor of 2. Song et al. (2023) distilled a consistency map that enables large time steps in the probability flow; Liu et al. (2022) retrained a flow-based method based on samples from a previously trained flow. Yang et al. (2023) used distillation to reduce model size while maintaining the quality of the generated images. The main drawbacks of distillation methods is their long training time (Salimans & Ho, 2022), and lack of consistency, *i.e.*, they do not sample from the distribution of the pre-trained model.

## 4 EXPERIMENTS

**Models and datasets.** Our method works with pre-trained models: we use the pre-trained CIFAR10 (Krizhevsky & Hinton, 2009) model of (Song et al., 2020b) with published weights from EDM (Karras et al., 2022). Additionally, we trained diffusion/flow models on the datasets: CI-

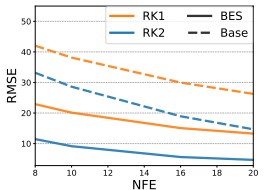
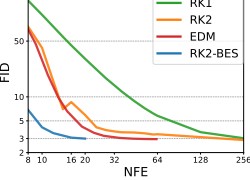

**Figure 3:** Bespoke RK1/2, ImageNet-64 FM-OT.

**Figure 4:** Bespoke solver applied to EDM's (Karras et al., 2022) CIFAR10 published model.

FAR10, AFHQ-256 (Choi et al., 2020a) and ImageNet-64/128 (Deng et al., 2009). Specifically, for ImageNet, as recommended by the authors (ima) we used the official *face-blurred* data ($64 \times 64$ downsampled using the open source preprocessing scripts from Chrabaszcz et al. (2017)). For diffusion models, we used an $\epsilon$-Variance Preserving ($\epsilon$-VP) parameterization and schedule (Ho et al., 2020; Song et al., 2020b). For flow models, we used Flow Matching (Lipman et al., 2022) with Conditional Optimal Transport (FM-OT), and Flow Matching/$v$-prediction with Cosine Scheduling (FM/$v$-CS) (Salimans & Ho, 2022; Albergo & Vanden-Eijnden, 2022). Note that Flow Matching methods directly provide the velocity vector field $u_t(x)$, and we converted $\epsilon$-VP to a velocity field using the identity in Song et al. (2020b). For conditional sampling we apply classifier free guidance (Ho & Salimans, 2022), so each evaluation uses two forward passes.

**Bespoke hyper-parameters and optimization.** As our base ODE solvers, we tested RK1 (Euler) and RK2 (Midpoint). Furthermore, we have two hyper-parameters $n$ – the number of steps, and $L_\tau$ – the Lipschitz constant from lemmas D.2, D.3. We train our models with $n \in \{4, 5, 8, 10, 12\}$ steps and fix $L_\tau = 1$. Ground Truth (GT) sample trajectories, $x(t_i)$, are computed with an adaptive RK45 solver (Shampine, 1986). We compute FID (Heusel et al., 2017) and validation RMSE (equation 6) is computed on a set of 10K fresh noise samples $x_0 \sim p(x_0)$; Figure 12 depicts an example of RMSE vs. training iterations for different $n$ values. Unless otherwise stated, below we report results on best FID iteration and show samples on best RMSE validation iteration. Figures 21, 22, 23 depict the learned Bespoke solvers' parameters $\theta$ for the experiments presented below; note the differences across the learned schemes for different models and datasets.

| | Method | | NFE | FID |
|---|---|---|---|---|
| **Distillation** | Zheng et al. (2023) | | 1 | 3.78 |
| | Luhman & Luhman (2021) | | 1 | 9.36 |
| | Salimans & Ho (2022) | | 1 | 9.12 |
| | | | 2 | 4.51 |
| | | | 8 | 2.57 |
| **Dedicated solvers** | DDIM(Song et al., 2020a) | | 10 | 13.36 |
| | | | 20 | 6.84 |
| | DPM (Lu et al., 2022a) | | 10 | 4.7 |
| | | | 20 | 3.99 |
| | DEIS (Zhang & Chen, 2022) | | 10 | 4.17 |
| | | | 20 | 2.86 |
| | GENIE (Dockhorn et al., 2022) | | 10 | 5.28 |
| | | | 20 | 3.94 |
| | DDSS (Watson et al., 2021) | | 10 | 7.86 |
| | | | 20 | 4.72 |
| | *RK2-BES* | $\epsilon$-VP | 10 | 3.31 |
| | | $\epsilon$-VP | 20 | 2.75 |
| | *RK2-BES* | FM/$v$-CS | 10 | 2.89 |
| | | FM/$v$-CS | 20 | 2.64 |
| | *RK2-BES* | FM-OT | 10 | **2.73** |
| | | FM-OT | 20 | **2.59** |

**Table 1:** CIFAR10 sampling.

**Bespoke RK1 vs. RK2.** We compared RK1 and RK2 and their Bespoke versions on CIFAR10 and ImageNet-64 models (FM-OT and FM/$v$-CS). Figure 3 and Figures 10, 9 show best validation RMSE (and corresponding PSNR). Using the same budget of function evaluations RK2/RK2-Bespoke produce considerably lower RMSE validation compared to RK1/RK1-Bespoke, respectively. We therefore opted for RK2/RK2-Bespoke for the rest of the experiments below.

**CIFAR10.** We tested our method on the pre-trained CIFAR10 $\epsilon$-VP model (Song et al., 2020b) released by EDM (Karras et al., 2022). Figure 4 compares our RK2-Bespoke solver to the EDM method, which corresponds to a particular choice of scaling, $s_i$, and time step discretization, $t_i$. Euler and EDM curves computed as originally implemented in EDM, where the latter achieves

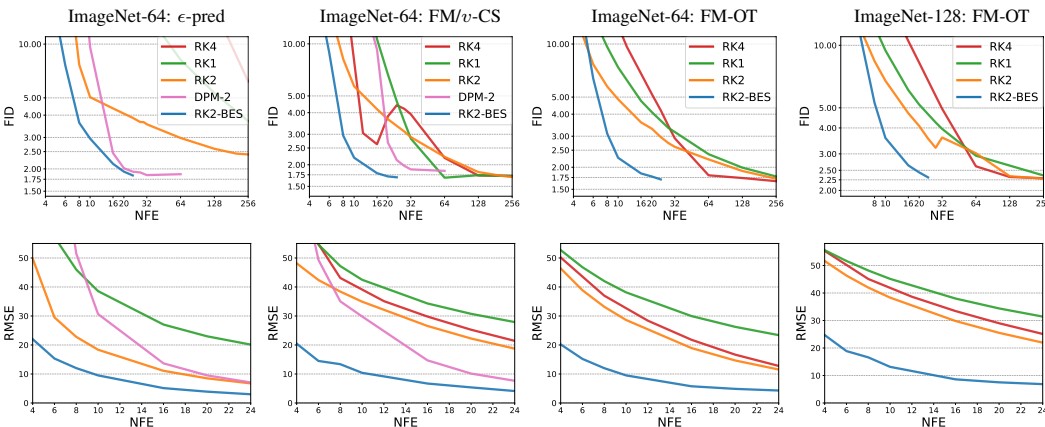

**Figure 5:** Bespoke RK2 solvers vs. RK1/2/4 solvers on CIFAR-10 ImageNet-64, and Image-Net128: FID vs. NFE (top row), and RMSE vs. NFE (bottom row). PSNR vs. NFE is shown in Figure 13.

FID=3.05 at 35 NFE, comparable to the result reported by EDM. Using our RK2-Bespoke Solver, we achieved an FID of 2.99 with 20 NFE, providing a 42% reduction in NFE. Additionally, we tested our method on three models we trained ourselves on CIFAR10, namely $\epsilon$-VP, FM/$v$-CS, and FM-OT. Table 1 compares our best FID for each model with different baselines demonstrating superior generation quality for low NFE among all dedicated solvers; *e.g.*, for NFE=10 we improve the FID of the runner-up by over 34% (from 4.17 to 2.73) using RK2-Bespoke FM-OT model. Table 4 lists best FID values for different NFE, along with the GT FID for the model and the fraction of time Bespoke training took compared to the original model's training time; with 20 NFE, our RK2-Bespoke solvers achieved FID within 8%, 1%, 1% (resp.) of the GT solvers' FID. Although close, our Bespoke solver does not match distillation's performance, however our approach is much faster to train, requiring ~1% of the original GPU training time with our naive implementation that re-samples the model at each iteration. Figure 11 shows FID/RMSE/PSNR vs. NFE, where PSNR is computed w.r.t. the GT solver's samples.

**ImageNet 64/128.** We further experimented with the more challenging ImageNet-64×64 / 128×128 datasets. For ImageNet-64 we also trained 3 models as described above. For ImageNet-128, due to computational budget constraints, we only trained FM-OT (training requires nearly 2000 GPU days). Figure 5 compares RK2-Bespoke to various baselines including DPM 2nd order (Lu et al., 2022a). As can be seen in the graphs, the Bespoke solvers improve both FID and RMSE. Interestingly, the Bespoke sampling takes all methods to similar RMSE levels, a fact that can be partially explained by Theorem 2.3. In Table 2, similar to Table 4, we report best FID per NFE for the Bespoke solvers we trained, the GT FID of the model, the % from GT achieved by the Bespoke solver, and the fraction of GPU time (in %) it took to train this Bespoke solver compared to training the original pre-trained model. Lastly, Figures 6, 7, 27, 28, 29, 25, 26 depict qualitative sampling examples for RK2-Bespoke and RK2 solvers. Note the significant improvement of fidelity in the Bespoke samples to the ground truth.

| ImageNet-64 | | NFE | FID | GT-FID/% | %Time |
|---|---|---|---|---|---|
| **RK2-BES** | $\epsilon$-VP | 8 | 3.63 | 1.83 / 229 | 3.5 |
| | $\epsilon$-VP | 10 | 2.96 | 163 | 3.6 |
| | $\epsilon$-VP | 16 | 2.14 | 120 | 3.6 |
| | $\epsilon$-VP | 20 | 1.93 | 109 | 3.5 |
| | $\epsilon$-VP | 24 | 1.84 | 101 | 3.6 |
| **RK2-BES** | FM/$v$-CS | 8 | 2.95 | 1.68 / 176 | 1.4 |
| | FM/$v$-CS | 10 | 2.20 | 131 | 1.6 |
| | FM/$v$-CS | 16 | 1.79 | 107 | 1.8 |
| | FM/$v$-CS | 20 | 1.71 | 102 | 1.5 |
| | FM/$v$-CS | 24 | 1.69 | 101 | 2.0 |
| **RK2-BES** | FM-OT | 8 | 3.10 | 1.68 / 185 | 1.6 |
| | FM-OT | 10 | 2.26 | 135 | 1.6 |
| | FM-OT | 16 | 1.84 | 110 | 1.7 |
| | FM-OT | 20 | 1.77 | 105 | 1.7 |
| | FM-OT | 24 | 1.71 | 102 | 1.8 |
| ImageNet-128 | | NFE | FID | GT-FID/% | %Time |
| **RK2-BES** | FM-OT | 8 | 5.28 | 2.30 / 230 | 1.1 |
| | FM-OT | 10 | 3.58 | 156 | 1.1 |
| | FM-OT | 16 | 2.64 | 115 | 1.2 |
| | FM-OT | 20 | 2.45 | 107 | 1.2 |
| | FM-OT | 24 | 2.31 | 101 | 1.2 |

**Table 2:** ImageNet Bespoke solvers.

**AFHQ-256.** We tested our method on the AFHQ dataset (Choi et al., 2020b) resized to 256×256 where as pre-trained model we used a FM-OT model we trained as described above. Figure 14 depicts PSNR/RMSE curves for the RK2-Bespoke solvers and baselines, and Figures 7 and 24 show qualitative sampling examples for RK2-Bespoke and RK2 solvers. Notice the high fidelity of the Bespoke generation samples.

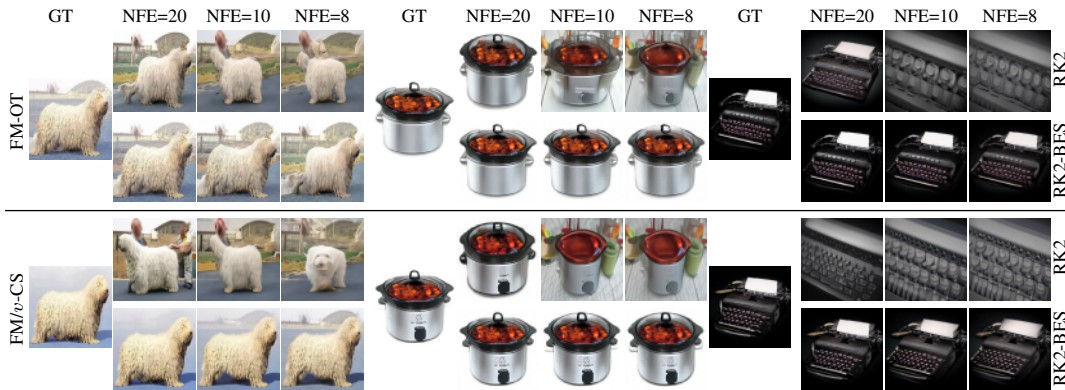

**Figure 6:** Comparison of FM-OT and FM/$v$-CS ImageNet-64 samples with RK2 and bespoke-RK2 solvers. Comparison to DPM-2 samples are in Figure 30. More examples are in Figures 27, 28, and 29. The similarity of generated images across models can be explained by their identical noise-to-data coupling (Theorem 2.3).

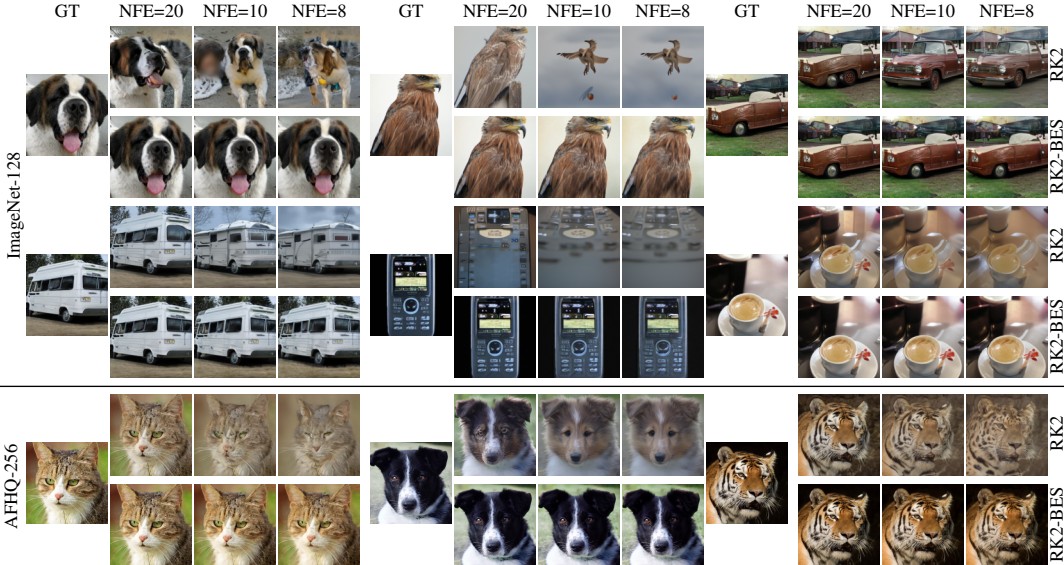

**Figure 7:** FM-OT ImageNet-128 (top) and AFHQ-256 (bottom) samples with RK2 and bespoke-RK2 solvers. More examples are in Figures 25, 26 and 24.

**Ablations.** We conducted two ablation experiments. First, Figure 16 shows the effect of training only time transform (keeping $s_r \equiv 1$) and scale transformation (keeping $t_r = r$). While the time transform is more significant than scale transform, incorporating scale improves RMSE for low NFE (which aligns with Theorem 2.2), and improve FID. Second, Figure 18 shows application of RK2-Bespoke solver trained on ImageNet-64 applied to ImageNet-128. The transferred solver, while sub-optimal compared to the Bespoke solver, still considerably improves the RK2 baseline in RMSE, and improves FID for higher NFE (16,20). Reusing Bespoke solvers can potentially be a cheap option to improve solvers.

## 5 CONCLUSIONS, LIMITATIONS AND FUTURE WORK

This paper develops an algorithm for finding low-NFE ODE solvers custom-tailored to general pre-trained flow models. Through extensive experiments we found that different models can benefit greatly from their own optimized solvers in terms of global truncation error (RMSE) and generation quality (FID). Currently, training a Bespoke solver requires roughly 1% of the original model's training time, which can probably be still be made more efficient, *e.g.*, by using training data or pre-processing sampling paths. A limitation of our framework is that it requires separate training for each target NFE and/or choice of guidance weight. For general NFE solvers one may consider a combined loss and/or continuous representation of $\varphi_r, t_r$, while guidance weight or even more general conditions can be used to directly condition $\varphi_r, t_r$. More general/expressive models for $\varphi_r, t_r$ have the potential to further improve fast sampling of pre-trained models.

ACKNOWLEDGEMENTS

NS is supported by a grant from Israel CHE Program for Data Science Research Centers.

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

## A   TRANSFORMED PATHS

(Appendix to Section 2.1.)

**Proposition A.1.** *Let $x(t)$ be a solution to equation 1. Denote $\dot{\varphi}_r := \frac{d}{dr}\varphi_r$ and $\dot{t}_r := \frac{d}{dr}t_r$. Then $\bar{x}(r)$ defined in equation 7 is a solution to the ODE (equation 1) with the VF*

$$\bar{u}_r(x) = \dot{\varphi}_r(\varphi_r^{-1}(x)) + \dot{t}_r\partial_x\varphi_r(\varphi_r^{-1}(x))u_{t_r}(\varphi_r^{-1}(x)). \tag{9}$$

*Proof.* Differentiating $\bar{x}(r)$ in equation 7, *i.e.*, $\bar{x}(r) = \varphi_r(x(t_r))$ w.r.t. $r$ and using the chain rule gives

$$\begin{aligned}
\dot{\bar{x}}(r) &= \frac{d}{dr}(\varphi_r(x(t_r))) \\
&= \dot{\varphi}_r(x(t_r)) + \partial_x\varphi_r(x(t_r))\dot{x}(t_r)\dot{t}_r \\
&= \dot{\varphi}_r(x(t_r)) + \partial_x\varphi_r(x(t_r))u_{t_r}(x(t_r))\dot{t}_r \\
&= \dot{\varphi}_r(\varphi_r^{-1}(\bar{x}(r))) + \partial_x\varphi_r(\varphi_r^{-1}(\bar{x}(r)))u_{t_r}(\varphi_r^{-1}(\bar{x}(r)))\dot{t}_r
\end{aligned}$$

where in the third equality we used the fact that $x(t)$ solves the ODE in equation 1 and therefore $\dot{x}(t) = u_t(x(t))$; and in the last equality we applied $\varphi_r^{-1}$ to both sides of equation 7, *i.e.*, $x(t_r) = \varphi_r^{-1}(\bar{x}(r))$. The above equation shows that

$$\dot{\bar{x}}(r) = u_r(\bar{x}(r)), \tag{28}$$

where $\bar{u}_r(x)$ is defined in equation 9, as required.   □

## B   CONSISTENCY OF SOLVERS

(Appendix to Section 2.1.)

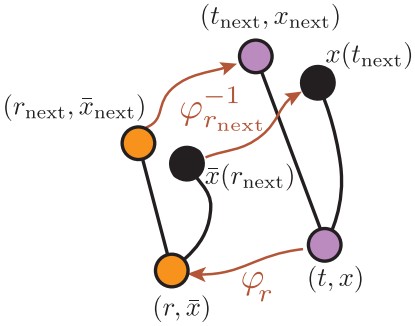

**Figure 8:** Proof notations and setup.

**Theorem 2.2.** *(Consistency of parametric solvers) Given arbitrary $t_r$, $\varphi_r$ in the family of functions $\mathcal{F}$ and a base ODE solver of order $k$, the corresponding ODE solver $\text{step}^\theta$ is also of order $k$, i.e.,*

$$\|x(t_{next}) - x_{next}\| = O((t_{next} - t)^{k+1}). \tag{13}$$

*Proof.* Here $(t, x)$ is our input sample $x \in \mathbb{R}^d$ at time $t \in [0, 1]$. By definition $r = r_t$, $r_{next} = r + h$, and $t_{next} = t_{r_{next}}$. Furthermore, by definition $\bar{x} = \varphi_r(x)$ is a sample at time $r$; $\bar{x}(r_{next})$ is the solution to the ODE defined by $\bar{u}_r$ starting from $(r, \bar{x})$; $\bar{x}_{next}$ is an approximation to $\bar{x}(r_{next})$ as generated from the base ODE solver step. Lastly, $x_{next} = \varphi_{r_{next}}^{-1}(\bar{x}_{r_{next}})$ and $x(t_{next}) = \varphi_{r_{next}}^{-1}(\bar{x}(r_{next}))$. See Figure 8 for an illustration visualizing this setup.

Now, since step is of order $k$ we have that

$$\begin{aligned}
\bar{x}(r_{next}) - \bar{x}_{next} &= \bar{x}(r_{next}) - \text{step}(\bar{x}, r; \bar{u}_r) \\
&= O((r_{next} - r)^{k+1}). \tag{29}
\end{aligned}$$

Now,

$$
\begin{aligned}
x(t_{\text{next}}) - x_{\text{next}} &= x(t_{\text{next}}) - \varphi_{r_{\text{next}}}^{-1}(\bar{x}_{\text{next}}) \\
&= x(t_{\text{next}}) - \varphi_{r_{\text{next}}}^{-1}(\bar{x}(r_{\text{next}}) + O((r_{\text{next}} - r)^{k+1})) \\
&= x(t_{\text{next}}) - \varphi_{r_{\text{next}}}^{-1}(\bar{x}(r_{\text{next}})) + O((r_{\text{next}} - r)^{k+1}) \\
&= O((r_{\text{next}} - r)^{k+1}) \\
&= O((t_{\text{next}} - t)^{k+1}),
\end{aligned}
$$

where in the first equality we used the definition of $x_{\text{next}}$; in the second equality we used equation 29; in the third equality we used the fact that $\varphi_r^{-1}$ is Lipschitz with constant $L$ (for all $r$); in the fourth equality we used the definition of the path transform, $x(t_{\text{next}}) = \varphi_{r_{\text{next}}}^{-1}(\bar{x}(r_{\text{next}}))$ as mentioned above; and in the last equality we used the fact that $r_t$ is also Lipschitz with a constant $L$ and therefore $r_{\text{next}} - r = r_{t_{\text{next}}} - r_t = O(t_{\text{next}} - t)$. $\qquad\square$

## C  EQUIVALENCE OF GAUSSIAN PATHS AND SCALE-TIME TRANSFORMATIONS

(Appendix to Section 2.2.)

**Theorem 2.3.** *(Equivalence of Gaussian Paths and scale-time transformation) Consider a Gaussian Path defined by a scheduler $(\alpha_t, \sigma_t)$, and let $x(t)$ denote the solution of equation 1 with $u_t$ defined in equation 23 and initial condition $x(0) = x_0$. Then,*

  (i) *For every other Gaussian Path defined by a scheduler $(\bar{\alpha}_r, \bar{\sigma}_r)$ with trajectories $\bar{x}(r)$ there exists a scale-time transformation with $s_1 = 1$ such that $\bar{x}(r) = s_r x(t_r)$.*

  (ii) *For every scale-time transformation with $s_1 = 1$ there exists a Gaussian Path defined by a scheduler $(\bar{\alpha}_r, \bar{\sigma}_r)$ with trajectories $\bar{x}(r)$ such that $s_r x(t_r) = \bar{x}(r)$.*

*Proof of theorem 2.3.* Consider two arbitrary schedulers $(\alpha_t, \sigma_t)$ and $(\bar{\alpha}_r, \bar{\sigma}_r)$. We can find $s_r, t_r$ such that

$$
\bar{\alpha}_r = s_r \alpha_{t_r}, \qquad \bar{\sigma}_r = s_r \sigma_{t_r}. \tag{30}
$$

Indeed, one can check the following are such $s_r, t_r$:

$$
t_r = \text{snr}^{-1}(\overline{\text{snr}}(r)), \qquad s_r = \frac{\bar{\sigma}_r}{\sigma_{t_r}}, \tag{31}
$$

where we remember snr is strictly monotonic as defined in equation 22, hence invertible. On the other hand, given an arbitrary scheduler $(\alpha_t, \sigma_t)$ and an arbitrary scale-time transformation $(t_r, s_r)$ with $s_1 = 1$, we can define a scheduler $(\bar{\alpha}_r, \bar{\sigma}_r)$ via equation 30.

For case (i), we are given another scheduler $\bar{\alpha}_r, \bar{\sigma}_r$ and define a scale-time transformation $s_r, t_r$ with equation 31. For case (ii), we are given a scale-time transformation $s_r, t_r$ and define a scheduler $\bar{\alpha}_r, \bar{\sigma}_r$ by equation 30.

Now, the scheduler $\bar{\alpha}_r, \bar{\sigma}_r$ defines sampling paths $\bar{x}(r)$ given by the solution of the ODE in equation 1 with the marginal VF $\bar{u}_r^{(1)}(x)$ defined in equation 23, *i.e.*,

$$
\bar{u}_r^{(1)}(x) = \int \bar{u}_r(x|x_1) \frac{\bar{p}_r(x|x_1)q(x_1)}{\bar{p}_r(x)} dx_1, \tag{32}
$$

where $\bar{u}_r(x|x_1) = \frac{\dot{\bar{\sigma}}_r}{\bar{\sigma}_r} x + \left[ \dot{\bar{\alpha}}_r - \dot{\bar{\sigma}}_r \frac{\bar{\alpha}_r}{\bar{\sigma}_r} \right] x_1$.

The scale-time transformation $s_r, t_r$ gives rise to a second VF $\bar{u}_r^{(2)}(x)$ as in equation 16,

$$
\bar{u}_r^{(2)}(x) = \frac{\dot{s}_r}{s_r} x + \dot{t}_r s_r u_{t_r}\left( \frac{x}{s_r} \right), \tag{33}
$$

where $u_t$ is the VF defined by the scheduler $(\alpha_t, \sigma_t)$ and equation 23.

By uniqueness of ODE solutions, the theorem will be proved if we show that

$$\bar{u}_r^{(1)}(x) = \bar{u}_r^{(2)}(x), \quad \forall x \in \mathbb{R}^d, r \in [0, 1]. \tag{34}$$

For that end, we use the notation of determinants to express

$$\bar{u}_r(x|x_1) = \frac{1}{\bar{\sigma}_r} \begin{vmatrix} 0 & x & x_1 \\ \bar{\sigma}_r & \bar{\alpha}_r & 1 \\ \dot{\bar{\sigma}}_r & \dot{\bar{\alpha}}_r & 0 \end{vmatrix}, \tag{35}$$

where $x$, $x_1 \in \mathbb{R}^d$ and $\bar{\alpha}_r, \bar{\sigma}_r, \dot{\bar{\alpha}}_r, \dot{\bar{\sigma}}_r \in \mathbb{R}$ as in vector cross product. Differentiating $\bar{\alpha}_r, \bar{\sigma}_r$ w.r.t. $r$ gives

$$\dot{\bar{\alpha}}_r = \dot{s}_r \alpha_{t_r} + s_r \dot{\alpha}_{t_r} \dot{t}_r, \qquad \dot{\bar{\sigma}}_r = \dot{s}_r \sigma_{t_r} + s_r \dot{\sigma}_{t_r} \dot{t}_r. \tag{36}$$

Using the bi-linearity of determinants shows that:

$$
\begin{aligned}
\bar{u}_r(x|x_1) &= \frac{1}{\bar{\sigma}_r} \begin{vmatrix} 0 & x & x_1 \\ \bar{\sigma}_r & \bar{\alpha}_r & 1 \\ \dot{\bar{\sigma}}_r & \dot{\bar{\alpha}}_r & 0 \end{vmatrix} \\
&= \frac{1}{s_r \sigma_{t_r}} \begin{vmatrix} 0 & x & x_1 \\ s_r \sigma_{t_r} & s_r \alpha_{t_r} & 1 \\ \dot{s}_r \sigma_{t_r} + s_r \dot{\sigma}_{t_r} \dot{t}_r & \dot{s}_r \alpha_{t_r} + s_r \dot{\alpha}_{t_r} \dot{t}_r & 0 \end{vmatrix} \\
&= \frac{1}{s_r \sigma_{t_r}} \begin{vmatrix} 0 & x & x_1 \\ s_r \sigma_{t_r} & s_r \alpha_{t_r} & 1 \\ \dot{s}_r \sigma_{t_r} & \dot{s}_r \alpha_{t_r} & 0 \end{vmatrix} + \frac{1}{s_r \sigma_{t_r}} \begin{vmatrix} 0 & x & x_1 \\ s_r \sigma_{t_r} & s_r \alpha_{t_r} & 1 \\ s_r \dot{\sigma}_{t_r} \dot{t}_r & s_r \dot{\alpha}_{t_r} \dot{t}_r & 0 \end{vmatrix} \\
&= \frac{\dot{s}_r}{s_r} x + \frac{s_r \dot{t}_r}{\sigma_{t_r}} \begin{vmatrix} 0 & \frac{x}{s_r} & x_1 \\ \sigma_{t_r} & \alpha_{t_r} & 1 \\ \dot{\sigma}_{t_r} & \dot{\alpha}_{t_r} & 0 \end{vmatrix} \\
&= \frac{\dot{s}_r}{s_r} x + s_r \dot{t}_r u_{t_r} \left( \frac{x}{s_r} \Big| x_1 \right),
\end{aligned}
$$

where in the second equality we substitute $\dot{\bar{\sigma}}_r, \dot{\bar{\alpha}}_r$ as in equation 36, in the third and fourth equality we used the bi-linearity of determinants, and in the last equality we used the definition of $u_t(x|x_1) = \frac{\dot{\sigma}_t}{\sigma_t} x + \left[ \dot{\alpha}_t - \dot{\sigma}_t \frac{\alpha_t}{\sigma_t} \right] x_1$ expressed in determinants notation. Furthermore, since

$$\bar{p}_r(x|x_1) = \mathcal{N}(x|s_r \alpha_{t_r} x_1, s_r^2 \sigma_{t_r}^2 I) \propto \mathcal{N}\left( \frac{x}{s_r} \Big| \alpha_{t_r} x_1, \sigma_{t_r}^2 I \right) = p_{t_r} \left( \frac{x}{s_r} \Big| x_1 \right) \tag{37}$$

we have that

$$\bar{p}_r(x_1|x) = p_{t_r} \left( x_1 \Big| \frac{x}{s_r} \right). \tag{38}$$

Therefore,

$$
\begin{aligned}
\int \bar{u}_r(x|x_1) \frac{\bar{p}_r(x|x_1) q(x_1)}{\bar{p}_r(x)} dx_1 &= \mathbb{E}_{\bar{p}_r(x_1|x)} \bar{u}_r(x|x_1) \\
&= \mathbb{E}_{p_{t_r}\left(x_1|\frac{x}{s_r}\right)} \left[ \frac{\dot{s}_r}{s_r} x + s_r \dot{t}_r u_{t_r} \left( \frac{x}{s_r} \Big| x_1 \right) \right] \\
&= \frac{\dot{s}_r}{s_r} x + s_r \dot{t}_r \mathbb{E}_{p_{t_r}\left(x_1|\frac{x}{s_r}\right)} u_{t_r} \left( \frac{x}{s_r} \Big| x_1 \right) \\
&= \frac{\dot{s}_r}{s_r} x + s_r \dot{t}_r u_{t_r} \left( \frac{x}{s_r} \right),
\end{aligned}
$$

where in the first equality we used Bayes rule, in the second equality we substitute $\bar{u}_r(x|x_1)$ and $\bar{p}_r(x_1|x)$ as above, and in the last equality we used the definition of $u_t$ as in equation 23. We have proved equation 34 and that concludes the proof. $\qquad \square$

# D  LIPSCHITZ CONSTANTS OF STEP$^\theta$.

(Appendix to Section 2.3.)

We are interested in computing $L_i^\theta$, a Lipschitz constant of the bespoke solver step function $\mathrm{step}_x^\theta(t_i, \cdot \, ; u_t)$. Namely, $L_i^\theta$ should satisfy

$$\left\| \mathrm{step}_x^\theta(t_i, x \, ; u_t) - \mathrm{step}_x^\theta(t_i, y \, ; u_t) \right\| \leq L_i^\theta \left\| x - y \right\|, \qquad \forall x, y \in \mathbb{R}^d. \tag{39}$$

We remember that $\mathrm{step}_x^\theta(t_i, \cdot \, ; u_t)$ is defined using a base solver and the VF $\bar{u}_{r_i}(\cdot)$; hence, we begin by computing a Lipschitz constant for $\bar{u}_{r_i}$ denoted $L_{\bar{u}}(r_i)$ in an auxiliary lemma:

**Lemma D.1.** *Assume that the original velocity field $u_t$ has a Lipschitz constant $L_u > 0$. Then for every $r_i \in [0, 1]$, $L_\tau \geq L_u$, and $x, y \in \mathbb{R}^d$*

$$\left\| \bar{u}_{r_i}(x) - \bar{u}_{r_i}(y) \right\| \leq L_{\bar{u}}(r_i) \left\| x - y \right\|, \tag{40}$$

*where*

$$L_{\bar{u}}(r_i) = \frac{|\dot{s}_i|}{s_i} + \dot{t}_i L_\tau \tag{41}$$

*Proof of lemma D.1.* Since the original velocity field $u$ has a Lipshitz constant $L_u > 0$, for every $t \in [0, 1]$ and $x, y \in \mathbb{R}^d$

$$\left\| u_t(x) - u_t(y) \right\| \leq L_u \left\| x - y \right\|. \tag{42}$$

Hence

$$\left\| \bar{u}_{r_i}(x) - \bar{u}_{r_i}(y) \right\| = \left\| \frac{\dot{s}_i}{s_i} x + \dot{t}_i s_i u_{t_i}\left( \frac{x}{s_i} \right) - \left( \frac{\dot{s}_i}{s_i} y + \dot{t}_i s_i u_{t_i}\left( \frac{y}{s_i} \right) \right) \right\| \tag{43}$$

$$= \left\| \frac{\dot{s}_i}{s_i}(x - y) + \dot{t}_i s_i \left( u_{t_i}\left( \frac{x}{s_i} \right) - u_{t_i}\left( \frac{y}{s_i} \right) \right) \right\| \tag{44}$$

$$\leq \frac{|\dot{s}_i|}{s_i} \left\| x - y \right\| + \dot{t}_i s_i \left\| u_{t_i}\left( \frac{x}{s_i} \right) - u_{t_i}\left( \frac{y}{s_i} \right) \right\| \tag{45}$$

$$\leq \left( \frac{|\dot{s}_i|}{s_i} + \dot{t}_i L_u \right) \left\| x - y \right\| \tag{46}$$

$$\leq \left( \frac{|\dot{s}_i|}{s_i} + \dot{t}_i L_\tau \right) \left\| x - y \right\|. \tag{47}$$

$\square$

We first apply the auxiliary lemma D.1 to compute a Lipschitz constant of $\mathrm{step}_x^\theta(t_i, \cdot \, ; u_t)$ with RK1 (Euler method) as the base solver in lemma D.2 and for RK2 (Midpoint method) as the base solver in lemma D.3.

**Lemma D.2.** *(RK1 Lipschitz constant) Assume that the original velocity field $u_t$ has a Lipschitz constant $L_u > 0$. Then, for every $L_\tau \geq L_u$,*

$$L_i^\theta = \frac{s_i}{s_{i+1}}\left( 1 + h L_{\bar{u}}(r_i) \right), \tag{48}$$

*is a Lipschitz constant of RK1-Bespoke update rule, where*

$$L_{\bar{u}}(r_i) = \frac{|\dot{s}_i|}{s_i} + \dot{t}_i L_\tau. \tag{49}$$

*Proof of lemma D.2.* We begin with writing an explicit expression of $\mathrm{step}_x^\theta(t_i, x, \, ; u_t)$ for Euler solver in terms of the transformed velocity field $\bar{u}_r$. That is,

$$\mathrm{step}_x^\theta(t_i, x, \, ; u_t) = \frac{1}{s_{i+1}}\left[ s_i x + h \bar{u}_{r_i}(s_i x) \right]. \tag{50}$$

So that applying the triangle inequality and lemma D.1 gives

$$
\begin{aligned}
\left\| \text{step}_x^\theta(t_i, x; u_t) - \text{step}_x^\theta(t_i, y; u_t) \right\| &= \frac{1}{s_{i+1}} \left\| s_i x + h \bar{u}_{r_i}(s_i x) - [s_i y + h \bar{u}_{r_i}(s_i y)] \right\| \\
&\leq \frac{s_i}{s_{i+1}} \|x - y\| + \frac{h}{s_{i+1}} \left\| \bar{u}_{r_i}(s_i x) - \bar{u}_{r_i}(s_i y) \right\| \\
&\leq \frac{s_i}{s_{i+1}} \|x - y\| + \frac{h}{s_{i+1}} \left( \frac{|\dot{s}_i|}{s_i} + \dot{t}_i L_\tau \right) \|s_i x - s_i y\| \\
&= \frac{s_i}{s_{i+1}} \left( 1 + h \left( \frac{|\dot{s}_i|}{s_i} + \dot{t}_i L_\tau \right) \right) \|x - y\|.
\end{aligned}
$$

$\square$

**Lemma D.3.** *(RK2 Lipschitz constant) Assume that the original velocity field $u_t$ has a Lipschitz constant $L_u > 0$. Then for every $L_\tau \geq L_u$*

$$
L_i^\theta = \frac{s_i}{s_{i+1}} \left[ 1 + h L_{\bar{u}}(r_{i+\frac{1}{2}}) \left( 1 + \frac{h}{2} L_{\bar{u}}(r_i) \right) \right] \tag{51}
$$

*is a Lipschitz constant of RK2-Bespoke update rule, where*

$$
L_{\bar{u}}(r_i) = \frac{|\dot{s}_i|}{s_i} + \dot{t}_i L_\tau. \tag{52}
$$

*Proof of lemma D.3.* We begin by writing explicit expression of $\text{step}_x^\theta(t_i, x; u_t)$ for RK2 (Midpoint) method in terms of the transformed velocity field $\bar{u}_r$. We set

$$
z = s_i x + \frac{h}{2} \bar{u}_{r_i}(s_i x), \quad w = s_i y + \frac{h}{2} \bar{u}_{r_i}(s_i y). \tag{53}
$$

then

$$
\text{step}_x^\theta(t_i, x, ; u_t) = \frac{1}{s_{i+1}} \left[ s_i x + h \bar{u}_{r_{i+\frac{1}{2}}}(z) \right], \tag{54}
$$

and

$$
\text{step}_x^\theta(t_i, y; u_t) = \frac{1}{s_{i+1}} \left[ s_i y + h \bar{u}_{r_{i+\frac{1}{2}}}(w) \right]. \tag{55}
$$

So that applying the triangle inequality and lemma D.1 gives

$$
\begin{aligned}
\left\| \text{step}_x^\theta(t_i, x; u_t) - \text{step}_x^\theta(t_i, y; u_t) \right\| &\leq \frac{s_i}{s_{i+1}} \|x - y\| + \frac{h}{s_{i+1}} \left\| \bar{u}_{r_{i+\frac{1}{2}}}(z) - \bar{u}_{r_{i+\frac{1}{2}}}(w) \right\| \\
&\leq \frac{s_i}{s_{i+1}} \|x - y\| + \frac{h}{s_{i+1}} L_{\bar{u}}(r_{i+\frac{1}{2}}) \|z - w\|. \tag{56}
\end{aligned}
$$

We apply the triangle inequality and the lemma D.1 again to $\|z - w\|$. That is,

$$
\begin{aligned}
\|z - w\| &= \left\| s_i x + \frac{h}{2} \bar{u}_{r_i}(s_i x) - \left( s_i y + \frac{h}{2} \bar{u}_{r_i}(s_i y) \right) \right\| \\
&\leq s_i \|x - y\| + \frac{h}{2} \left\| \bar{u}_{r_i}(s_i x) - \bar{u}_{r_i}(s_i y) \right\| \\
&\leq s_i \|x - y\| + \frac{h}{2} L_{\bar{u}}(r_i) s_i \|x - y\| \\
&= s_i \left( 1 + \frac{h}{2} L_{\bar{u}}(r_i) \right) \|x - y\|.
\end{aligned}
$$

Substitute back in equation 56 gives

$$
\frac{s_i}{s_{i+1}} \|x - y\| + \frac{h}{s_{i+1}} L_{\bar{u}}(r_{i+\frac{1}{2}}) \|z - w\| \leq \frac{s_i}{s_{i+1}} \left[ 1 + h L_{\bar{u}}(r_{i+\frac{1}{2}}) \left( 1 + \frac{h}{2} L_{\bar{u}}(r_i) \right) \right] \|x - y\|.
$$

$\square$

# E  DERIVATION OF PARAMETRIC SOLVER STEP$^\theta$

(Appendix to Section 2.2.)
This section presents a derivation of $n$-step parametric solver

$$\text{step}^\theta(t, x \,; u_t) = \left(\text{step}^\theta_t(t, x \,; u_t), \text{step}^\theta_x(t, x \,; u_t)\right) \tag{57}$$

for scale-time transformation (equation 15) with two options for a base solver: (i) RK1 method (Euler) as the base solver; and $(ii)$ RK2 method (Midpoint). We do so by following equation 10-12. We begin with RK1 and derive equation 17. Given $(t_i, x_i)$, equation 10 for the scale time transformation is,

$$\bar{x}_i = s_i x_i. \tag{58}$$

Then according to equation 11,

$$\bar{x}_{i+1} = \text{step}_x(r_i, \bar{x}_i, \bar{u}_{r_i}) \tag{59}$$

$$= \bar{x}_i + h \bar{u}_{r_i}(\bar{x}_i) \tag{60}$$

$$= \bar{x}_i + h \left( \frac{\dot{s}_i}{s_i} \bar{x}_i + \dot{t}_i s_i u_{t_i} \left( \frac{\bar{x}_i}{s_i} \right) \right) \tag{61}$$

$$= s_i x_i + h \left( \dot{s}_i x_i + \dot{t}_i s_i u_{t_i}(x_i) \right), \tag{62}$$

where in the second equality we apply an RK1 step (equation 4), in the third equality we substitute $\bar{u}_{r_i}$ using equation 16, and in the fourth equality we substitute $\bar{x}_i$ as in equation 58. According to RK1 step (equation 4) we also have $r_{i+1} = r_i + h$. Finally, equation 12 gives,

$$\text{step}^\theta_t(t_i, x_i \,; u_t) = t_{i+1} \tag{63}$$

$$\text{step}^\theta_x(t_i, x_i \,; u_t) = \frac{s_i + h \dot{s}_i}{s_{i+1}} x_i + \frac{h}{s_{i+1}} \dot{t}_i s_i u_{t_i}(x_i), \tag{64}$$

as in equation 17.

Regarding the second case, equation 11 for the RK2 method (equation 5) is,

$$\bar{x}_{i+1} = \text{step}_x(r_i, \bar{x}_i, \bar{u}_{r_i}) \tag{65}$$

$$= \bar{x}_i + h \bar{u}_{r_{i+\frac{1}{2}}}(\bar{x}_{i+\frac{1}{2}}), \tag{66}$$

where

$$\bar{x}_{i+\frac{1}{2}} = \bar{x}_i + \frac{h}{2} \bar{u}_{r_i}(\bar{x}_i) \tag{67}$$

is the RK1 step from $(r_i, \bar{x}_i)$ with step size $h/2$. Now substituting $\bar{x}_i$ as defined equation 58 and $\bar{u}_r$ as defined in equation 16 we get

$$\bar{x}_{i+1} = s_i x_i + h \left( \frac{\dot{s}_{i+\frac{1}{2}}}{s_{i+\frac{1}{2}}} \bar{x}_{i+\frac{1}{2}} + \dot{t}_{i+\frac{1}{2}} s_{i+\frac{1}{2}} u_{t_{i+\frac{1}{2}}} \left( \frac{\bar{x}_{i+\frac{1}{2}}}{s_{i+\frac{1}{2}}} \right) \right). \tag{68}$$

where

$$\bar{x}_{i+\frac{1}{2}} = \left( s_i + \frac{h}{2} \dot{s}_i \right) x_i + \frac{h}{2} s_i \dot{t}_i u_{t_i}(x_i). \tag{69}$$

Lastly, according to equation 12 we have

$$\text{step}^\theta_x(t_i, x \,; u_t) = \frac{s_i}{s_{i+1}} x_i + \frac{h}{s_{i+1}} \left( \frac{\dot{s}_{i+\frac{1}{2}}}{s_{i+\frac{1}{2}}} \bar{x}_{i+\frac{1}{2}} + \dot{t}_{i+\frac{1}{2}} s_{i+\frac{1}{2}} u_{t_{i+\frac{1}{2}}} \left( \frac{\bar{x}_{i+\frac{1}{2}}}{s_{i+\frac{1}{2}}} \right) \right), \tag{70}$$

as in equation 19 where $z_i = \bar{x}_{i+\frac{1}{2}}$.

# F    IMPLEMENTATION DETAILS

(Appendix to Section 2.3.)

This section presents further implementation details, complementing the main text. Our parametric family of solvers step$^\theta$ is defined via a base solver step and a transformation $(t_r, \varphi_r)$ as defined in equation 12. We consider the RK2 (Midpoint, equation 5) method as the base solver with $n$ steps and $(t_r, \varphi_r)$ the scale-time transformation (equation 15). That is, $\varphi_r(x) = s_r x$, where $s : [0, 1] \to \mathbb{R}_{>0}$, as in equation 14, which is our primary use case.

**Parameterization of $t_i$.** Remember that $t_r$ is a strictly monotonic, differentiable, increasing function $t : [0, 1] \to [0, 1]$. Hence, $t_i$ must satisfy the constraints as in equation 21, *i.e.*,

$$0 = t_0 < t_{\frac{1}{2}} < \cdots < t_n = 1 \tag{71}$$

$$\dot{t}_0, \dot{t}_{\frac{1}{2}}, \ldots, \dot{t}_{n-1}, \dot{t}_{n-\frac{1}{2}} > 0. \tag{72}$$

To satisfy these constrains, we model $t_i$ and $\dot{t}_i$ via

$$t_i = \frac{\sum_{j=0}^{i} |\theta_j^t|}{\sum_{k=0}^{n} |\theta_k^t|}, \quad \dot{t}_i = |\theta_i^{\dot{t}}|, \tag{73}$$

where $\theta_i^t$ and $\theta_i^{\dot{t}}$, $i = 0, \frac{1}{2}, ..., n$ are free learnable parameters.

**Parameterization of $s_i$.** Since $s_r$ is a strictly positive, differentiable function satisfying a boundary condition at $r = 0$, the sequence $s_i$ should satisfy the constraints as in equation 21, *i.e.*,

$$s_{\frac{1}{2}}, s_1, \ldots, s_n > 0 \;\; , \;\; s_0 = 1, \tag{74}$$

and $\dot{s}_i$ are unconstrained. Similar to the above, we model $s_i$ and $\dot{s}_i$ by

$$s_i = \begin{cases} 0 & i = 0 \\ \exp \theta_i^s & \text{otherwise} \end{cases}, \quad \dot{s}_i = \theta_i^{\dot{s}}, \tag{75}$$

where $\theta_i^s$ and $\theta_i^{\dot{s}}$, $i = 0, \frac{1}{2}, ..., n$ are free learnable parameters.

**Bespoke training.** The pseudo-code for training a Bespoke solver is provided in Algorithm 3. Here we add some more details on different steps of the training algorithm. We initialize the parameters $\theta$ such that the scale-transformation is the Identity transformation. That is, for every $i = 0, \frac{1}{2}, ..., n$,

$$t_i = \frac{i}{n}, \quad \dot{t}_i = 1, \tag{76}$$

$$s_i = 1, \quad \dot{s}_i = 0. \tag{77}$$

Explicitly, in terms of the learnable parameters, for every $i = 0, \frac{1}{2}, ..., n$,

$$\theta_i^t = 1, \quad \theta_i^{\dot{t}} = 1 \tag{78}$$

$$\theta_i^s = 0, \quad \theta_i^{\dot{s}} = 0. \tag{79}$$

To compute the GT path $x(t)$, we solve the ODE in equation 1 with the pre-trained model $u_t$ and DOPRI5 method, then use linear interpolation to extract $x(t_i)$, $i = 0, 1, ..., n$ (Chen, 2018). Then, apply $x_i^{\mathrm{aux}}(t)$ (equation 27) to correctly handle the gradients w.r.t. to $\theta_i^t$. See Table 3 for number of trajectories used during training. To compute the loss $\mathcal{L}_{\mathrm{RMSE\text{-}B}}$ (equation 26) we compute $x_{i+1} = \mathrm{step}_x^\theta \left( x_i^{\mathrm{aux}}(t_i), t_i; u_t \right)$ with equations 19,20, and compute $M_i$ via lemma D.3 with $L_\tau = 1$. Finally, we use Adam optimizer Kingma & Ba (2017) with a learning rate of $2e^{-3}$.

**Efficient sampling.** When sampling using a Bespoke solver (Algorithm 2) each step involves applying $\varphi_{r_i}^{-1}$ and $\varphi_{r_i}$ consecutively. In case we use scale transformation, equation 14 (as is done in all examples in this paper), this does not introduce any difficulty, however if a more compute intensive $\varphi$ is used the following sampling pseudo-code (Algorithm 4) provides an equivalent sampling while avoiding this unnecessary step.

|  | CIFAR10 | ImageNet-64 | ImageNet-128 | AFHQ 256 |
|---|---|---|---|---|
| Total number of trajectories | 72k | 48k | 48k | 4k |
| Batch size | 12 | 8 | 8 | 1 |
| Number of iterations | 6k | 6k | 6k | 4k |

**Table 3:** Hyper-parameters of Bespoke solvers training on CIFAR10/ImageNet-64/ImageNet-128/AFHQ 256.

---

**Algorithm 4** Bespoke sampling (efficient).

---

**Require:** pre-trained $u_t$, trained $\theta$
   $x_0 \sim p(x_0)$             ▷ sample noise
   $r_0 \leftarrow 0, \bar{x}_0 \leftarrow x_0$      ▷ initial conditions
   **for** $i = 0, 1, \ldots, n-1$ **do**
      $(r_{i+1}, \bar{x}_{i+1}) \leftarrow \text{step}(r_i, \bar{x}_i; \bar{u}_r^\theta)$
   **end for**
   **return** $\varphi_1^{-1}(\bar{x}_n)$

---

## G   BESPOKE RK1 VERSUS RK2

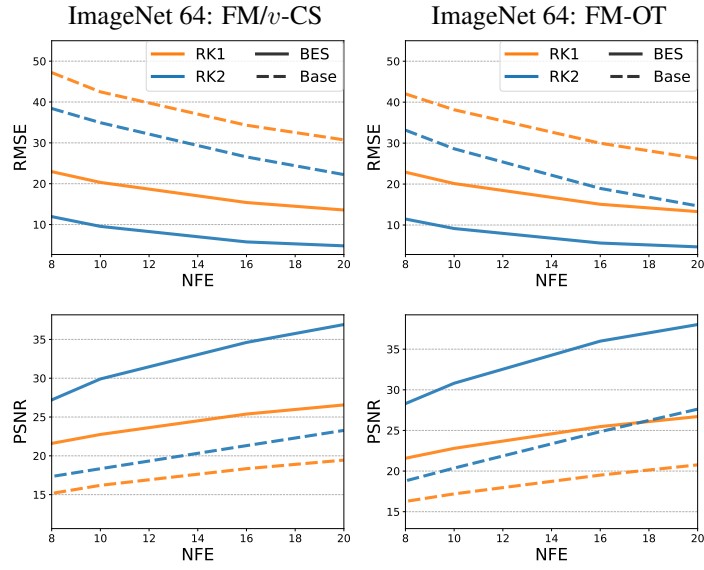

**Figure 9:** Bespoke RK1, Bespoke RK2, RK1, and RK2 solvers on ImageNet-64 models: RMSE vs. NFE (top row), and PSNR vs. NFE (bottom row).

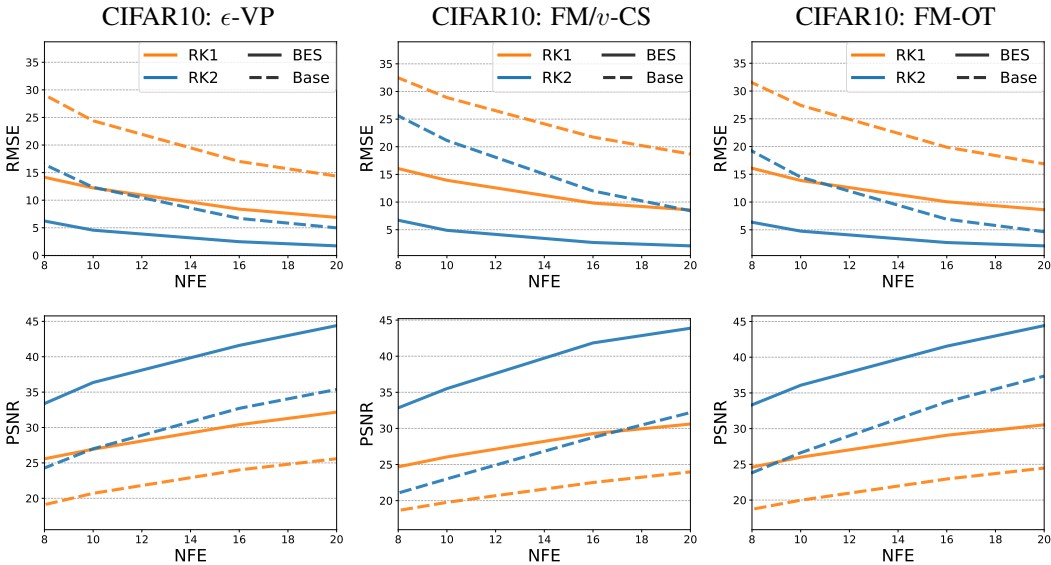

**Figure 10:** Bespoke RK1, Bespoke RK2, RK1, and RK2 solvers on CIFAR10: RMSE vs. NFE (top row), and PSNR vs. NFE (bottom row).

# H  CIFAR10

| CIFAR10 | | NFE | FID | GT-FID/% | | %Time |
|---------|---|-----|-----|------|-----|-------|
| ***RK2-BES*** | $\epsilon$-VP | 8 | 4.26 | 2.54 / | 168 | 1.4 |
| | $\epsilon$-VP | 10 | 3.31 | | 130 | 1.5 |
| | $\epsilon$-VP | 16 | 2.84 | | 112 | 1.5 |
| | $\epsilon$-VP | 20 | 2.75 | | 108 | 1.4 |
| ***RK2-BES*** | FM/$v$-CS | 8 | 3.50 | 2.61 / | 134 | 0.5 |
| | FM/$v$-CS | 10 | 2.89 | | 111 | 0.6 |
| | FM/$v$-CS | 16 | 2.68 | | 103 | 0.6 |
| | FM/$v$-CS | 20 | 2.64 | | 101 | 0.6 |
| ***RK2-BES*** | FM-OT | 8 | 3.13 | 2.57 / | 122 | 0.5 |
| | FM-OT | 10 | 2.73 | | 106 | 0.6 |
| | FM-OT | 16 | 2.60 | | 101 | 0.6 |
| | FM-OT | 20 | 2.59 | | 101 | 0.6 |

**Table 4:** CIFAR10 Bespoke solvers. We report best FID vs. NFE, the ground truth FID (GT-FID) for the model and FID/GT-FID in %, and the fraction of GPU time (in %) required to train the bespoke solver w.r.t. training the original model.

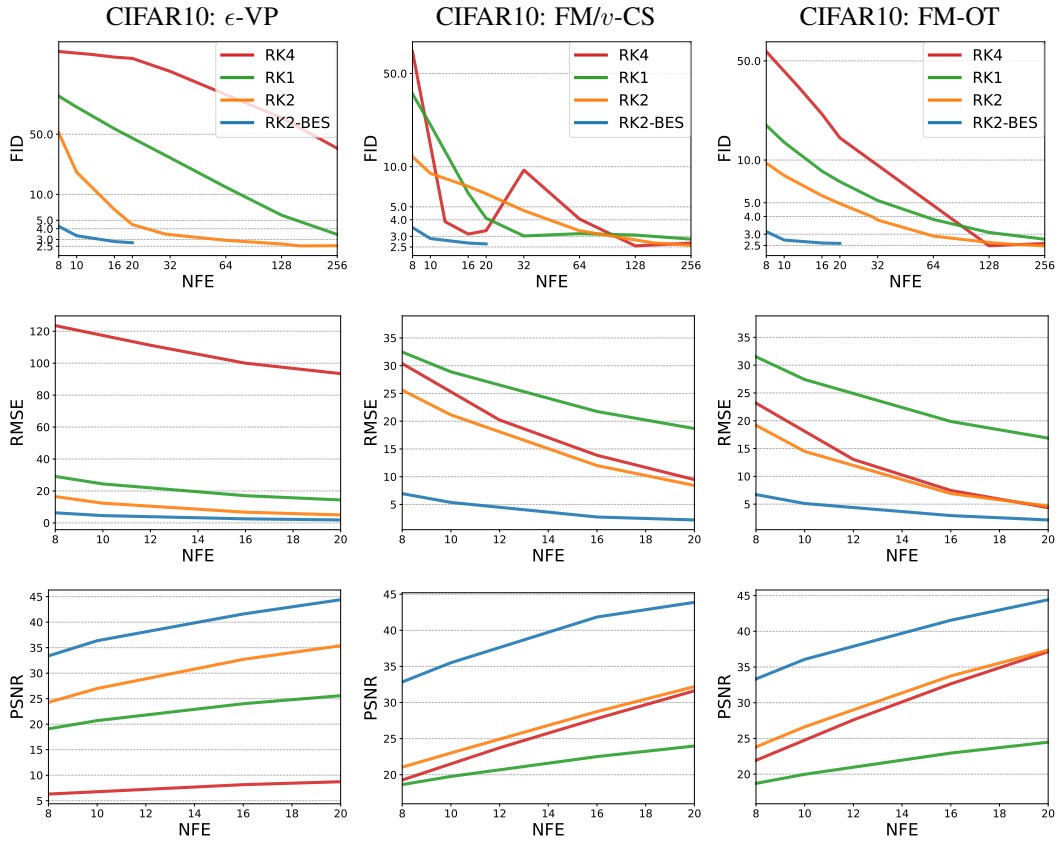

**Figure 11:** CIFAR10 sampling with Bespoke RK2 solvers vs. RK1,RK2,RK4: FID vs. NFE (top row), RMSE vs. NFE (middle row), and PSNR vs. NFE (bottom row).

# I   IMAGENET-64/128

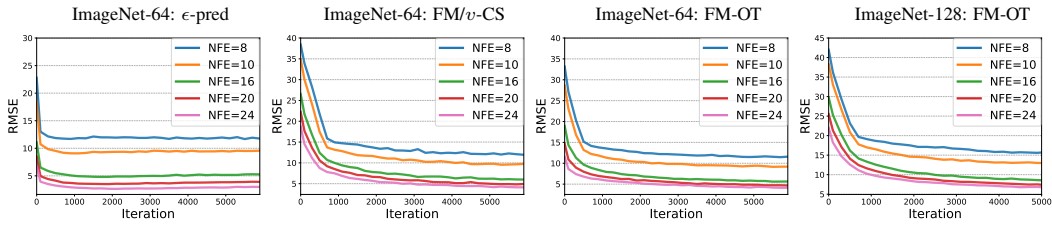

**Figure 12:** Validation RMSE vs. training iterations of Bespoke RK2 solvers on ImageNet-64, and ImageNet-128.

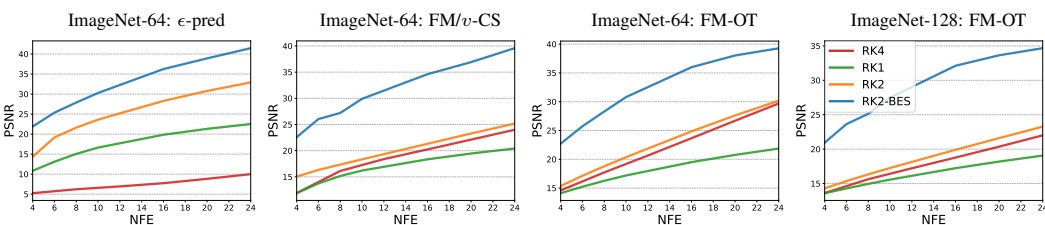

**Figure 13:** Bespoke RK2, RK1, RK2, and RK4 solvers on ImageNet-64, and ImageNet-128; PSNR vs. NFE.

## J AFHQ-256

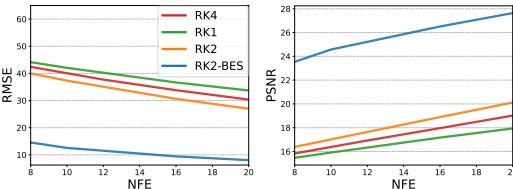

**Figure 14:** Bespoke RK2, RK1, RK2, and RK4 solvers on AFHQ-256; PSNR vs. NFE (left), and RMSE vs. NFE (right).

## K    ABLATIONS

### K.1    LOSS ABLATION

We consider here three losses for optimizing the Bespoke solvers: RMSE-Bound (the parallel loss we advocate in the paper), RMSE (optimizing directly equation 6), and a simplified version of the RMSE-Bound: sum of Local Truncation Errors (LTE-parallel). That is, LTE is defined as in equation 26 but taking $M_i^\theta = 1$ for all $i$. Algorithm 5 provides the pseudo-codes for all three losses. We have run all three algorithms on the ImageNet 64 dataset and compared their FID, and RMSE, where RMSE is computed w.r.t. GT samples (see Section 4 for details). For the non-parallel RMSE loss, we needed to use Activate Checkpointing to reduce memory consumption in order to be able to run this loss. Figure 15 shows the results. As can be seen in the graphs, RMSE loss, as expected, reaches lowest RMSE values per NFE, the second best is the RMSE-Bound loss and worst in terms of RMSE is the LTE. As for FID, RMSE performs worst, while for FM-OT model RMSE-Bound and LTE perform equivalently for NFE>10, and LTE has advantage as far as FID goes otherwise. Since our goal is to reduce RMSE and provide memory-scalable training algorithm we opted to use the memory efficient RMSE-Bound in the paper.

---

**Algorithm 5** Bespoke training (parallel).

---

**Require:**  pre-trained $u_t$, number of steps $n$
  initialize $\theta \in \mathbb{R}^p$
  **while** not converged **do**
    $x_0 \sim p(x_0)$                                                                       ▷ sample noise
    $x(t) \leftarrow$ solve ODE 1                                                              ▷ GT path
    **if** RMSE loss **then**
      $x_n^\theta \leftarrow$ Bespoke sampling                                                  ▷ Alg. 2
      $\mathcal{L} \leftarrow \left\| x(1) - x_n^\theta \right\|$
    **else if** RMSE-parallel loss **then**
      $\mathcal{L} \leftarrow 0$                                                            ▷ init loss
      **parallel for** $i = 0, ..., n-1$ **do**
        $x_{i+1}^\theta \leftarrow \mathrm{step}_x^\theta \left( x_i^{\mathrm{aux}}(t_i), t_i; u_t \right)$
        $\mathcal{L} \mathrel{+}= M_{i+1}^\theta \left\| x_{i+1}^{\mathrm{aux}}(t_{i+1}) - x_{i+1}^\theta \right\|$
      **end for**
    **else if** LTE-parallel loss **then**
      $\mathcal{L} \leftarrow 0$                                                            ▷ init loss
      **parallel for** $i = 0, ..., n-1$ **do**
        $x_{i+1}^\theta \leftarrow \mathrm{step}_x^\theta \left( x_i^{\mathrm{aux}}(t_i), t_i; u_t \right)$
        $\mathcal{L} \mathrel{+}= \left\| x_{i+1}^{\mathrm{aux}}(t_{i+1}) - x_{i+1}^\theta \right\|$
      **end for**
    **end if**
    $\theta \leftarrow \theta - \gamma \nabla_\theta \mathcal{L}$                               ▷ optimization step
  **end while**
  **return** $\theta$

---

### K.2    SCALE-TIME ABLATIONS

This section presents an ablation experiment on the effect of each component in the scale-time transformation. We train Bespoke-RK2 solvers with three choices of transformation: (i) time-only: train $t_r, \dot{t}_r$ and freeze $s_r \equiv 1, \dot{s}_r \equiv 0$, (ii) scale-only: freeze $t_r \equiv r, \dot{t}_r \equiv 1$ and train $s_r, \dot{s}_r$, and (iii) scale-time: train both $t_r, \dot{t}_r$ and $s_r, \dot{s}_r$. All experiments are performed on ImageNet 64 FM-OT model. Figure 16 shows the FID, and RMSE of Bespoke-RK2 time-only/scale-only/scale-time solvers and the base RK2 solver. First, all three Bespoke-RK2 solvers improve upon the base RK2 solver. Second, the time component seems more significant, but the scale component improves FID for all NFEs and RMSE for NFE $< 20$. Third, in RMSE, the significance of the time component increases as NFE increases. In addition, Figure 17 shows the trained time-only (top) and scale-only (bottom) transformations. Interestingly, we see that even seemingly small changes (e.g., scale-only with NFE $\in \{16, 20\}$) can affect dramatically the FID.

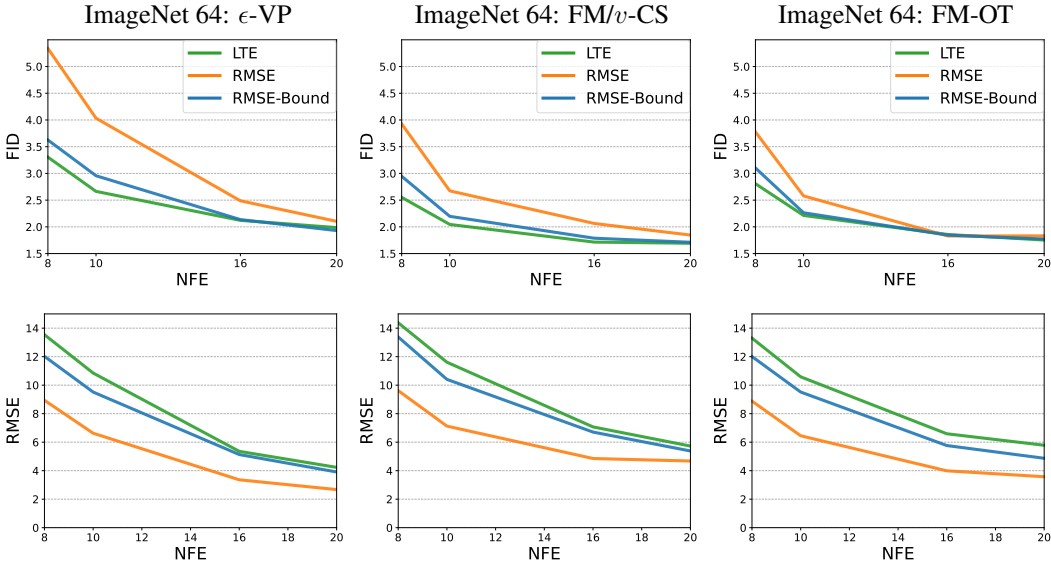

**Figure 15:** Different Bespoke losses for ImageNet 64. We compare the RMSE-Bound (the parallel loss advocated in the paper), direct RMSE loss, and Local Truncation Error (LTE) which is a slightly simplified version of RMSE-Bound loss. All three variations are provided in Algorithm 5. In terms of RMSE the direct RMSE loss is as expected best however has large memory footprint, while RMSE-Bound is the runner-up and parallelizable. FID is not perfectly correlated with RMSE and shows a somewhat opposite trend (partially excluding the FM-OT model where RMSE-Bound and LTE are almost FID equivalent). We opted for RMSE-Bound loss in the paper since it is memory-scalable and provides best RMSE among the parallel loss options considered.

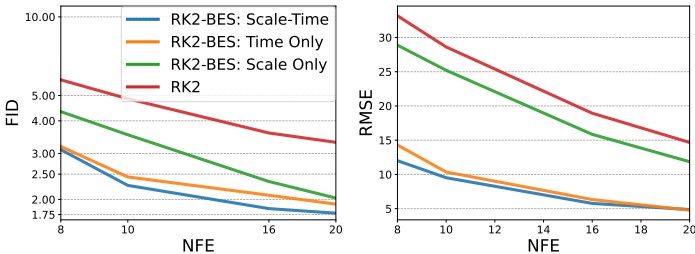

**Figure 16:** Bespoke ablation I: RK2, Bespoke RK2 with full scale-time optimization, time-only optimization (keeping $s_r \equiv 1$ fixed), and scale-only optimization (keeping $t_r = r$ fixed) on FM-OT ImageNet-64: FID vs. NFE (left), and RMSE vs. NFE (right). Note that most improvement provided by time optimization where scale improves FID for all NFEs, and RMSE for $< 20$ NFEs.

### K.3 TRANSFERRING BESPOKE SOLVERS

This section presents an ablation experiment demonstrating trained Bespoke solvers' generalization to different models. We train a Bespoke-RK2 solver on an ImageNet-64 FM-OT model and evaluate it on an ImageNet-128 FM-OT model. We compare its FID, and RMSE vs. NFE against the base RK2 solver evaluated on ImageNet-128 FM-OT and a Bespoke-RK2 solver trained and evaluated on ImageNet-128 FM-OT. The results is shown in Figure 18.

### K.4 DISTILLATION-TYPE PARAMETRIZATION

This section presents a distillation-type experiment in our framework. Our parametric family of solvers is defined by a composition of a scale-time transformation and the RK2 solver, resulting in step$^\theta$ as in equations 19 and 20, where the weights of the pre-trained model $u_t$ are frozen. A natural comparison to a distillation-like approach is to let the weights of $u_t = u_t^\theta$ change during

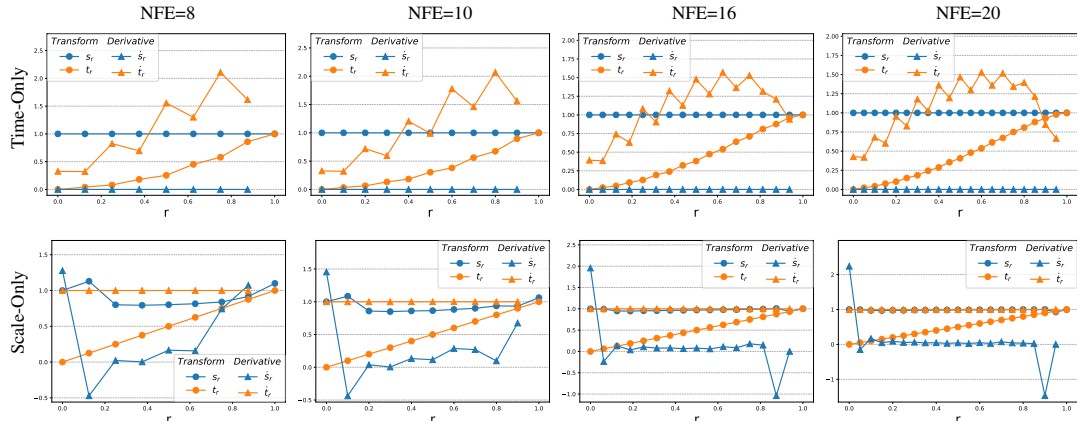

**Figure 17:** Trained $\theta$ of scale-time ablation: Bespoke-RK2 time-only optimization (top), Bespoke-RK2 scale-only optimization (bottom), on ImageNet-64 FM-OT for NFE 8/10/16/20.

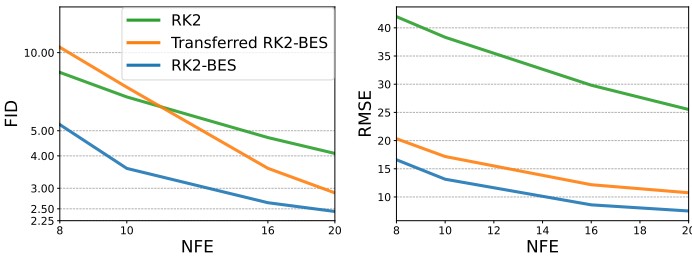

**Figure 18:** Bespoke ablation II: RK2 evaluated on FM-OT ImageNet-128 model, Bespoke RK2 trained and evaluated on FM-OT ImageNet-128 model, and Bespoke RK2 trained on FM-OT ImageNet-64 and evaluated on FM-OT ImageNet-128 model (transferred): FID vs. NFE (left), and RMSE vs. NFE (right). Note that the transferred Bespoke solver is still inferior to the Bespoke solver but improves considerably RMSE compared to the RK2 baseline. In FID the transferred solver improves over the baseline only for NFE=16,20.

optimization instead of using the scale-time transformation, that is

$$\mathcal{L}_{\text{dis}}(\theta) = \mathbb{E}_{x_0 \sim p(x_0)} \sum_{i=0}^{n-1} \left\| x_{i+1}(t_{i+1}) - \text{step}_x(x_i, t_i; u_t^\theta) \right\|, \tag{80}$$

where $t_i = ih$ is fixed. We perform this experiment on ImageNet 64 FM$v$-CS. For a fair comparison, we use the same compute budget as used to train our Bespoke-RK2 solvers on these models: a total of 48K generated trajectories, 6k iterations, and we report at best FID. Figure 19 shows the RK2 base solver, Bespoke-RK2 solver, and Distillation-RK2 on ImageNet 64 FM$v$-CS: FID vs. NFE (left), and RMSE vs. NFE (right). Note that while distillation is able to improve from the baseline solver (RK2) it does not match the performance of the Bespoke solver. Two potential explanations why distillation is not as performant as Bespoke in this experiment are: First, the amount of trajectories/compute budget we use to optimize Bespoke is not sufficient for effective distillation; and second, successful distillation methods require access to training data. Training only on generated data distillation is less effective.

## K.5 TRAINING STOPPING CRITERIA

In this ablation experiment, we qualitatively compare samples when changing the stopping criteria of the bespoke solver training. In Figure 20, we compare samples with bespoke-RK2 solvers where the training stopping criteria is best FID versus best RMSE on four models, ImageNet 64 $\epsilon$-VP/FM$v$-CS/FM-OT and ImageNet 128 FM-OT, and NFE $\in \{8, 10, 20\}$. In all of the cases the differences are practically indistinguishable.

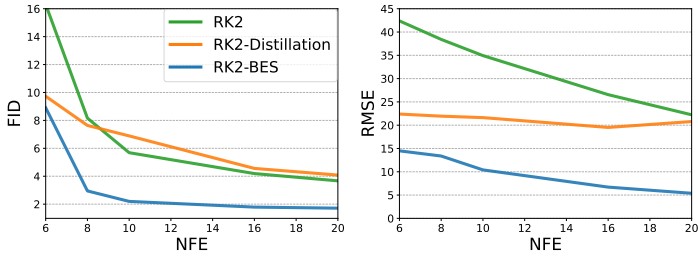

**Figure 19:** Distillation-type experiment: RK2 solver, Bespoke-R2 solver, and Distillation using same number of trajectories and compute as Bespoke, on ImageNet 64 FM/$v$-CS: FID vs. NFE (left), and RMSE vs. NFE (right).

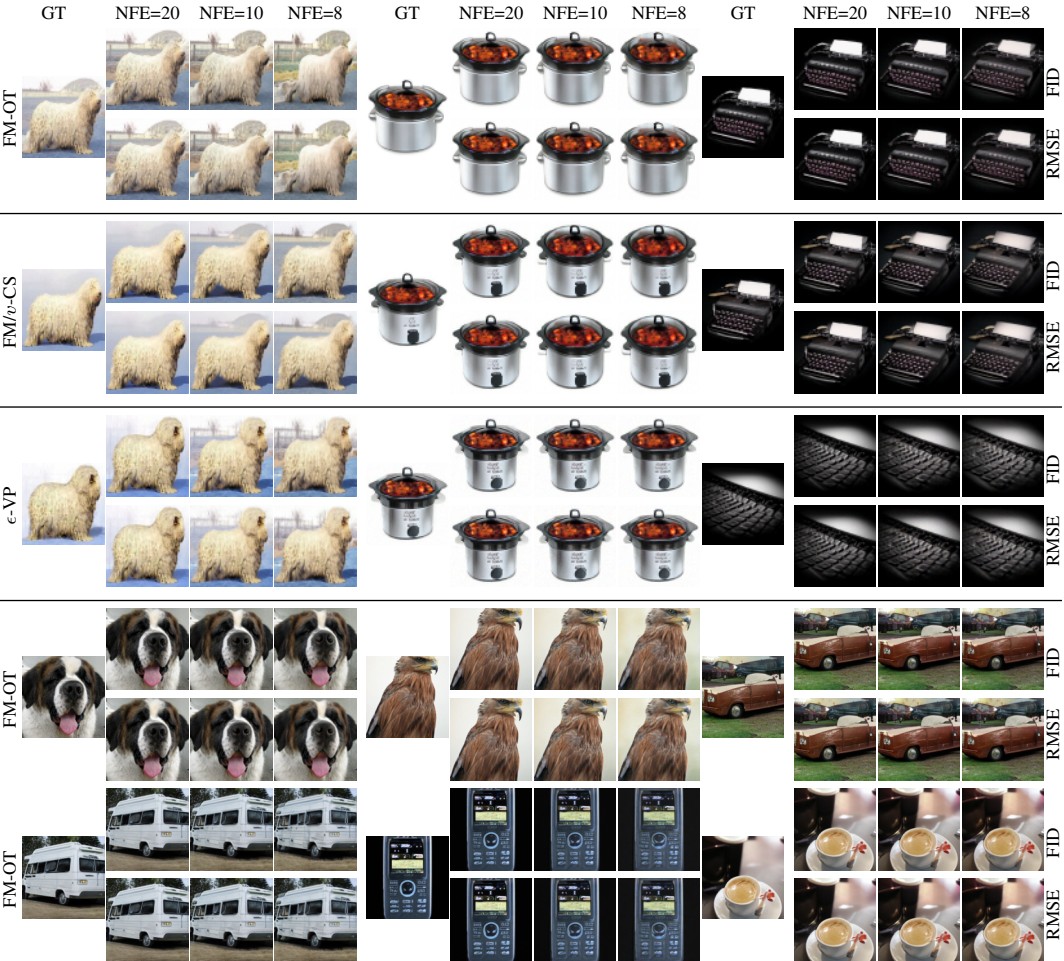

**Figure 20:** Different stopping criteria experiment. We compare samples with bespoke-RK2 solvers with training stopping criteria at best FID and best RMSE; ImageNet64 (3 top rows) and ImageNet-128 (2 bottom rows).

## L   TRAINED BESPOKE SOLVERS

In this section, we present the trained Bespoke solvers by visualizing their respective parameters $\theta$. Figures 21, 22, and 23 show the learned scale-time transformation of Bespoke-RK2 solvers trained on ImageNet-128, ImageNet-64, and CIFAR10 (resp.) for NFE $\{8, 10, 16, 20\}$. First, we note the significant differences between the learned scale-time transformation of $\epsilon$-VP versus FM-OT on ImageNet 64 in Figure 22 top and bottom rows (resp.). Second, we note that the scale-time transformations trained on the same model type but on different datasets seem to have similarities

to some degree but are still different from one another, see Figure 21 and Figures 22, 23 bottom rows, for FM-OT trained on ImageNet-128, ImageNet-64, and CIFAR10 (resp.). These two observations showcase the advantage of a custom-made solver for each model. We also tested the latter point empirically in the ablation experiment in Section K.3, where we tested a Bespoke-RK2 solver trained on an ImageNet 64 FM-OT model to an ImageNet 128 FM-OT model and noticed a drop in performance compared to a Bespoke solver trained directly on the ImageNet 128 FM-OT model. In addition, we note the resemblance in the form of scale-time transformations trained on the same model type and same dataset across different NFE (i.e., a row in Figures 21, 22, and 23). This phenomenon suggests there may be a well-defined scale-time transformation in the limit of NFE $\to \infty$. Furthermore, for $\dot{t}_i$ and $\dot{s}_i$, even and odd parity points on the grid seem to converge to different curves, possibly due to their different role in the RK2 solvers.

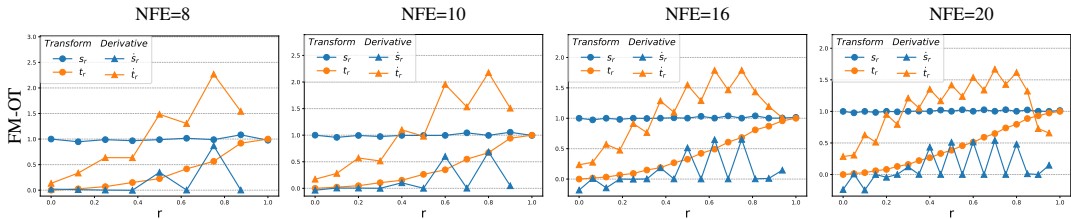

**Figure 21:** Trained $\theta$ of Bespoke-RK2 solvers on ImageNet-128 FM-OT for NFE 8/10/16/20.

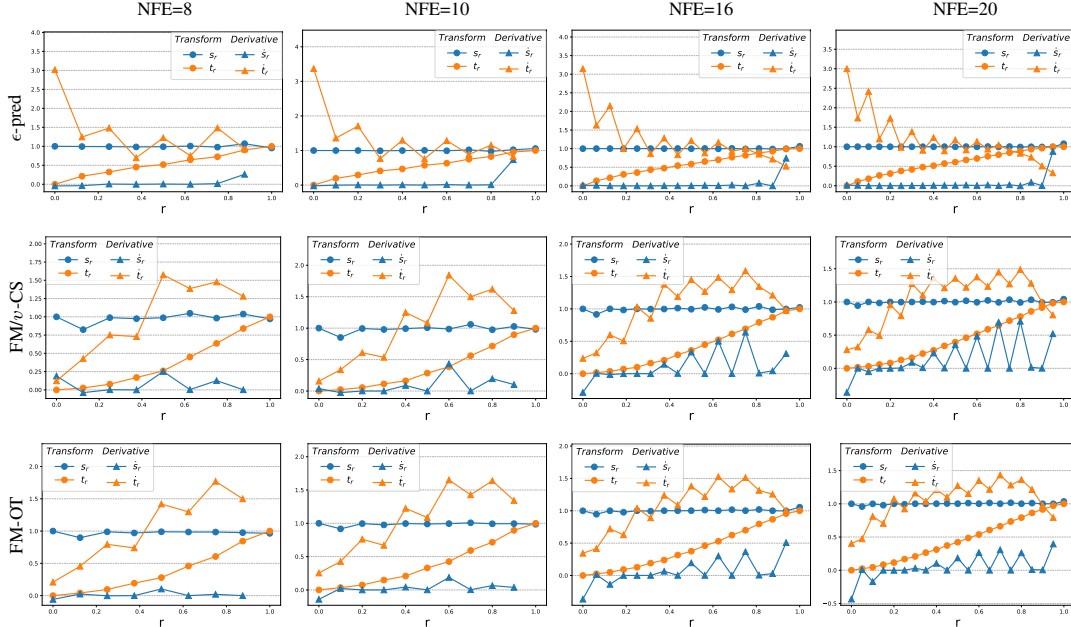

**Figure 22:** Trained $\theta$ of Bespoke-RK2 solvers on ImageNet-64 for NFE 8/10/16/20; $\epsilon$-VP (top), FM/$v$-CS (middle), and FM-OT (bottom).

## M PRE-TRAINED MODELS

All our FM-OT and FM/$v$-CS models were trained with Conditional Flow Matching (CFM) loss derived in Lipman et al. (2022),

$$\mathcal{L}_{\text{CFM}}(\theta) = \mathbb{E}_{t,p_0(x_0),q(x_1)} \left\| v_t(x_t;\theta) - (\dot{\sigma}_t x_0 + \dot{\alpha}_t x_1) \right\|^2, \tag{81}$$

where $t \sim \mathcal{U}([0,1])$, $p_0(x_0) = \mathcal{N}(x_0|0,I)$, $q(x_1)$ is the data distribution, $v_t(x_t;\theta)$ is the network, $(\alpha_t, \sigma_t)$ is the scheduler as defined in equation 22, and $x_t = \sigma_t x_0 + \alpha_t x_1$. For FM-OT the scheduler

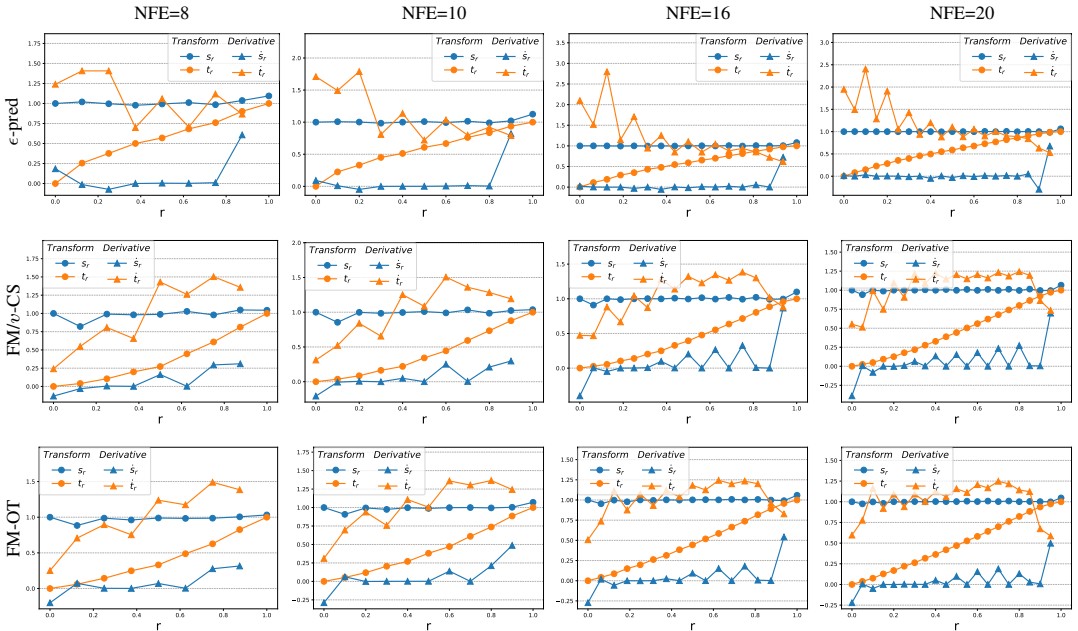

**Figure 23:** Trained $\theta$ of Bespoke-RK2 solvers on CIFAR10 for NFE 8/10/16/20; $\epsilon$-pred (top), FM/$v$-CS (middle), and FM-OT (bottom).

is

$$\alpha_t = t, \quad \sigma_t = 1 - t, \tag{82}$$

and for FM/$v$-CS the scheduler is

$$\alpha_t = \sin \frac{\pi}{2}t, \quad \sigma_t = \cos \frac{\pi}{2}t. \tag{83}$$

All our $\epsilon$-VP models were trained with noise prediction loss as derived in Ho et al. (2020) and Song et al. (2020b),

$$\mathcal{L}_{\text{noise}}(\theta) = \mathbb{E}_{t,p_0(x_0),q(x_1)} \left\| \epsilon_t(x_t; \theta) - x_0 \right\|^2, \tag{84}$$

where the VP scheduler is

$$\alpha_t = \xi_{1-t}, \quad \sigma_t = \sqrt{1 - \xi_{1-t}^2}, \quad \xi_s = e^{-\frac{1}{4}s^2(B-b) - \frac{1}{2}sb}, \tag{85}$$

and $B = 20$, $b = 0.1$. All models use U-Net architecture as in Dhariwal & Nichol (2021), and the hyper-parameters are listed in Table 5.

| | CIFAR10 $\epsilon$-VP | CIFAR10 FM-OT;FM/$v$-CS | ImageNet-64 $\epsilon$-VP;FM-OT;FM/$v$-CS | ImageNet-128 FM-OT | AFHQ 256 FM-OT |
|---|---|---|---|---|---|
| Channels | 128 | 128 | 196 | 256 | 256 |
| Depth | 4 | 4 | 3 | 2 | 2 |
| Channels multiple | 2,2,2 | 2,2,2 | 1,2,3,4 | 1,1,2,3,4 | 1,1,2,2,4,4 |
| Heads | 1 | 1 | - | - | - |
| Heads Channels | - | - | 64 | 64 | 64 |
| Attention resolution | 16 | 16 | 32,16,8 | 32,16,8 | 64,32,16 |
| Dropout | 0.1 | 0.3 | 1.0 | 0.0 | 0.0 |
| Effective Batch size | 512 | 512 | 2048 | 2048 | 256 |
| GPUs | 8 | 8 | 64 | 64 | 64 |
| Epochs | 2000 | 3000 | 1600 | 1437 | 862 |
| Iterations | 200k | 300k | 1M | 900k | 50k |
| Learning Rate | 5e-4 | 1e-4 | 1e-4 | 1e-4 | 1e-4 |
| Learning Rate Scheduler | constant | constant | constant | Poly Decay | Polyn Decay |
| Warmup Steps | - | - | - | 5k | 5k |
| P-Unconditional | - | - | 0.2 | 0.2 | 0.2 |
| Guidance weight | - | - | 0.20 (vp,cs), 0.15 (ot) | 0.5 | 0.1 |
| Total parameters count | 55M | 55M | 296M | 421M | 537M |

**Table 5:** Pre-trained models' hyper-parameters.

# N    MORE RESULTS

In this section we present more sampling results using RK2-Bespoke solvers, the RK2 baseline and Ground Truth samples (with DOPRI5).

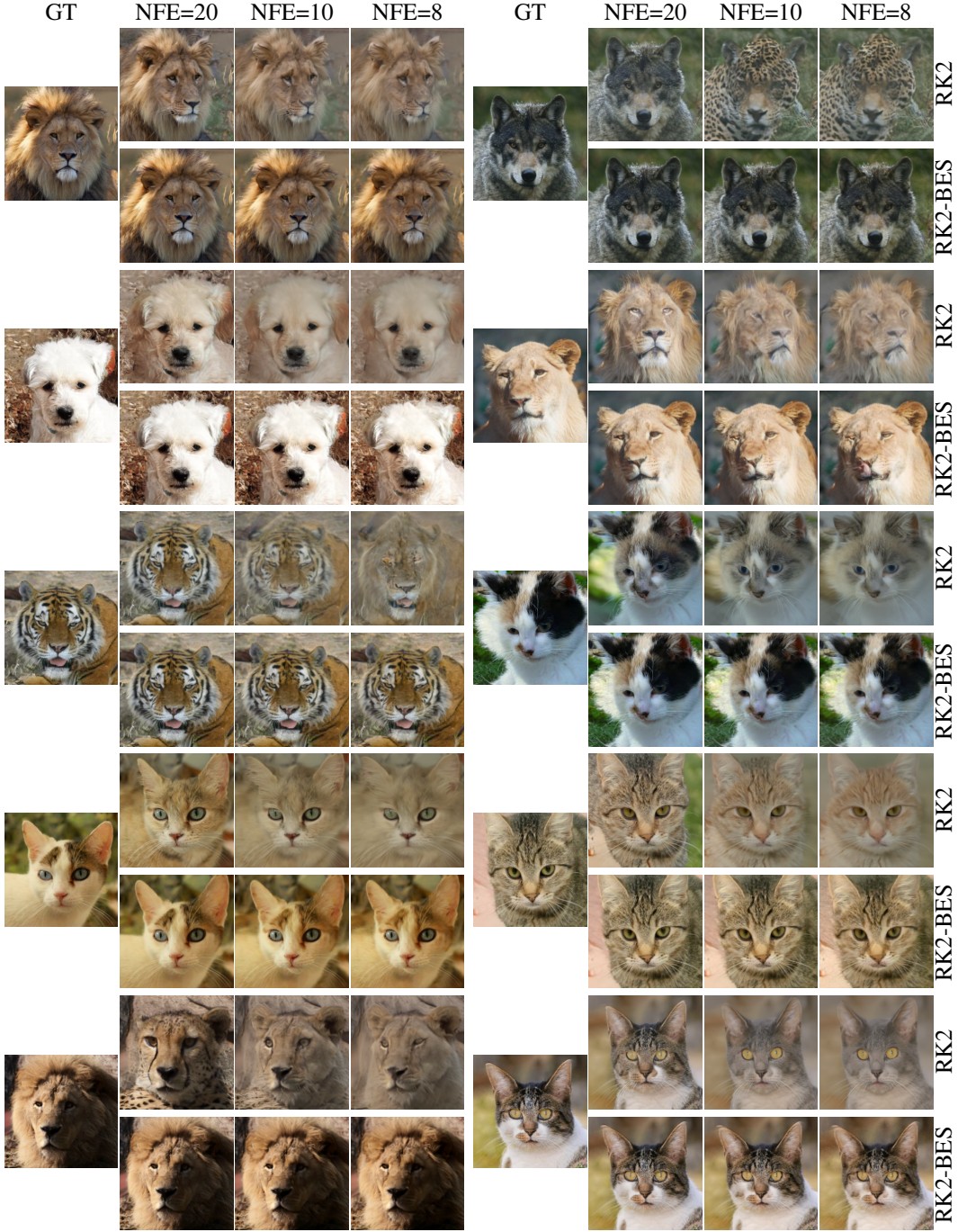

**Figure 24:** Comparison of FM-OT AFHQ-256 GT samples with RK2 and Bespoke-RK2 solvers.

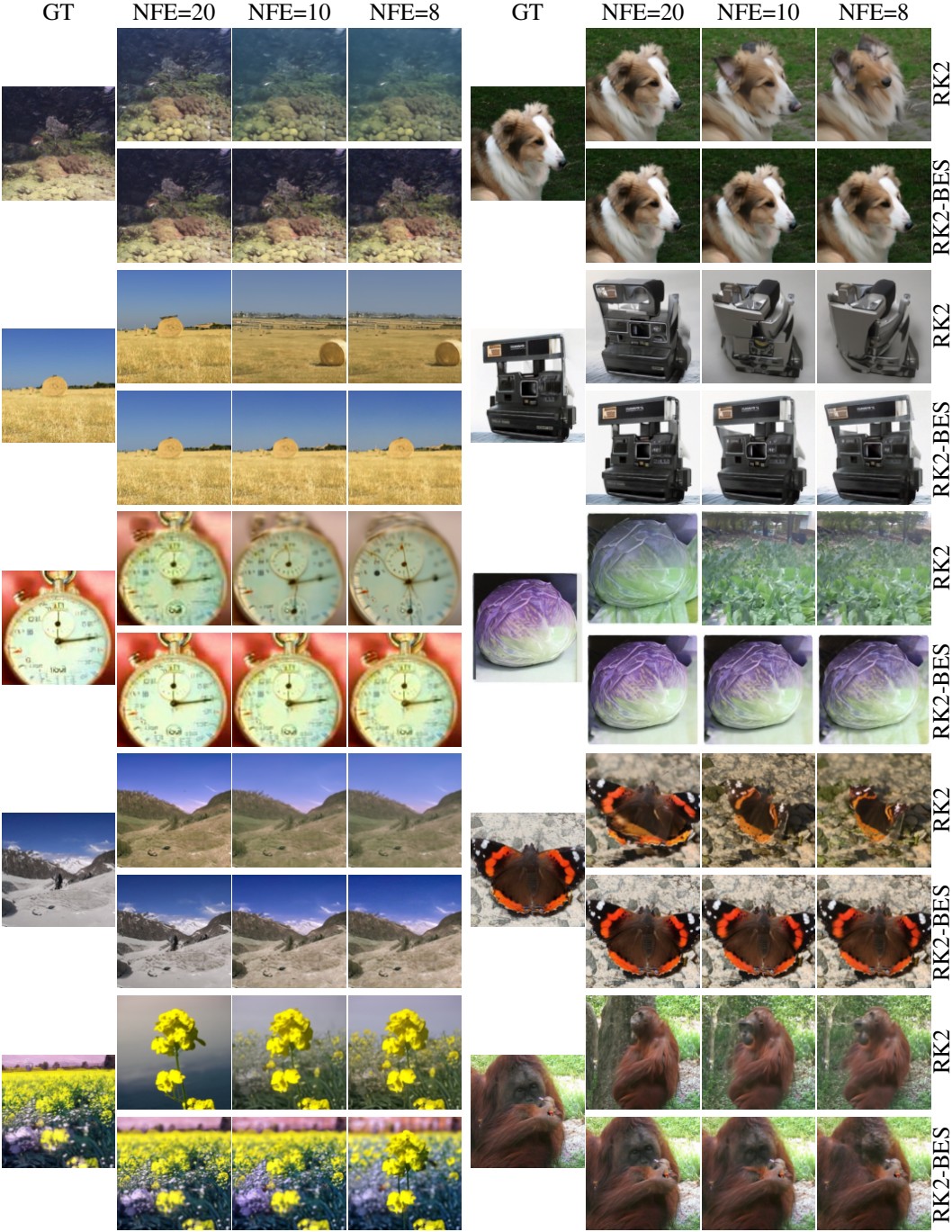

**Figure 25:** Comparison of FM-OT ImageNet-128 GT samples with RK2 and Bespoke-RK2 solvers.

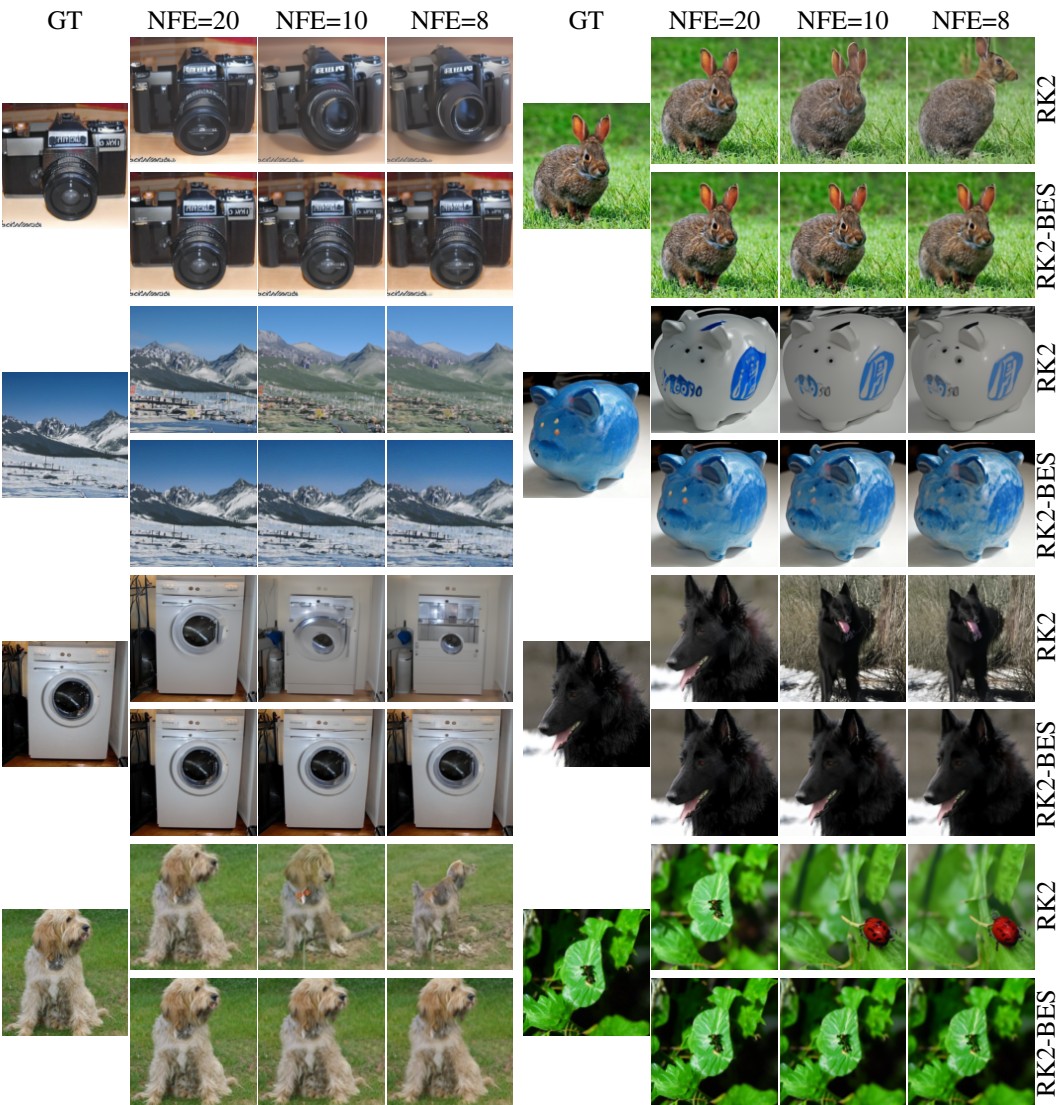

**Figure 26:** Comparison of FM-OT ImageNet-128 GT samples with RK2 and Bespoke-RK2 solvers.

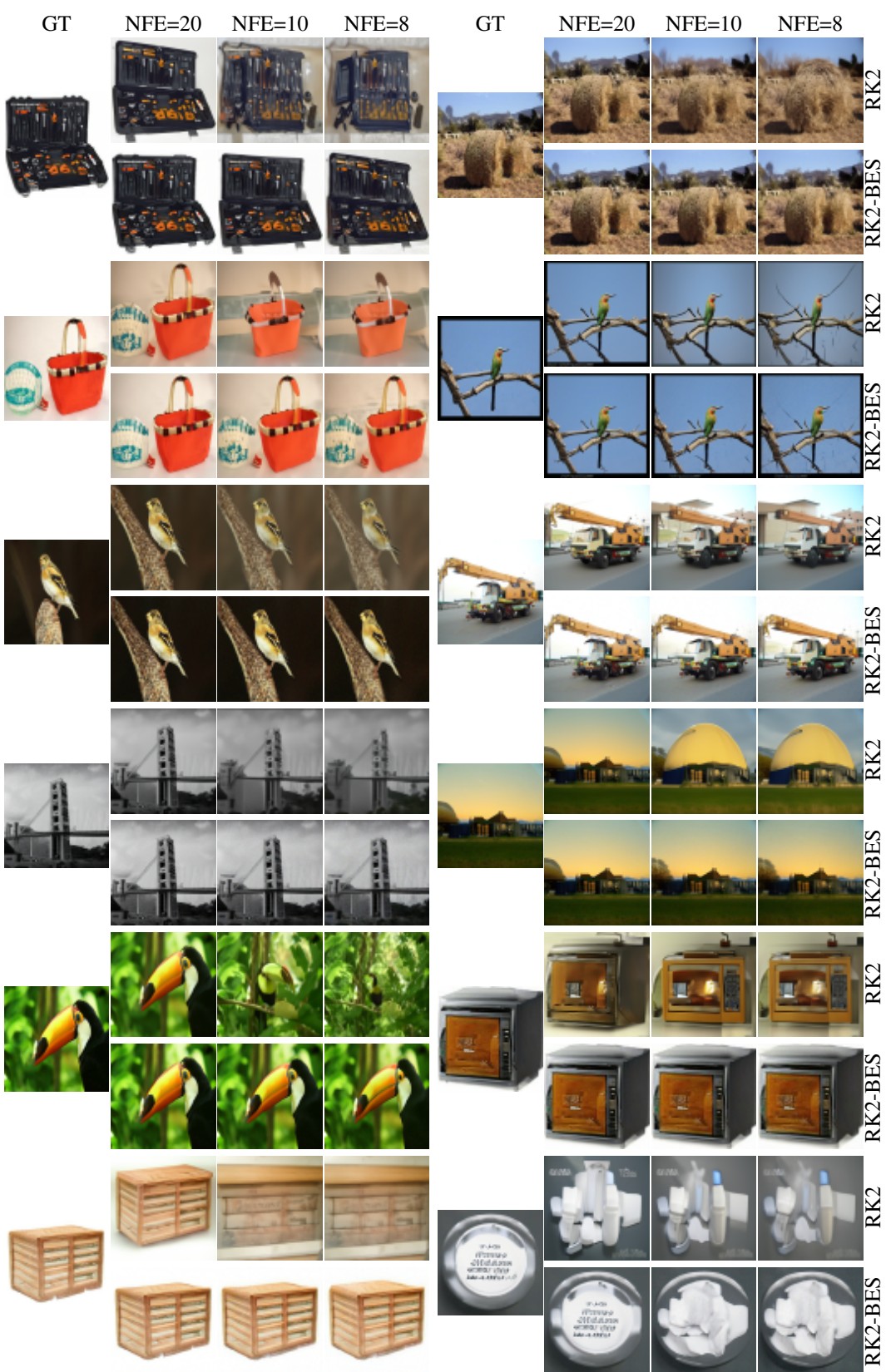

**Figure 27:** Comparison of FM-OT ImageNet-64 GT samples with RK2 and Bespoke-RK2 solvers.

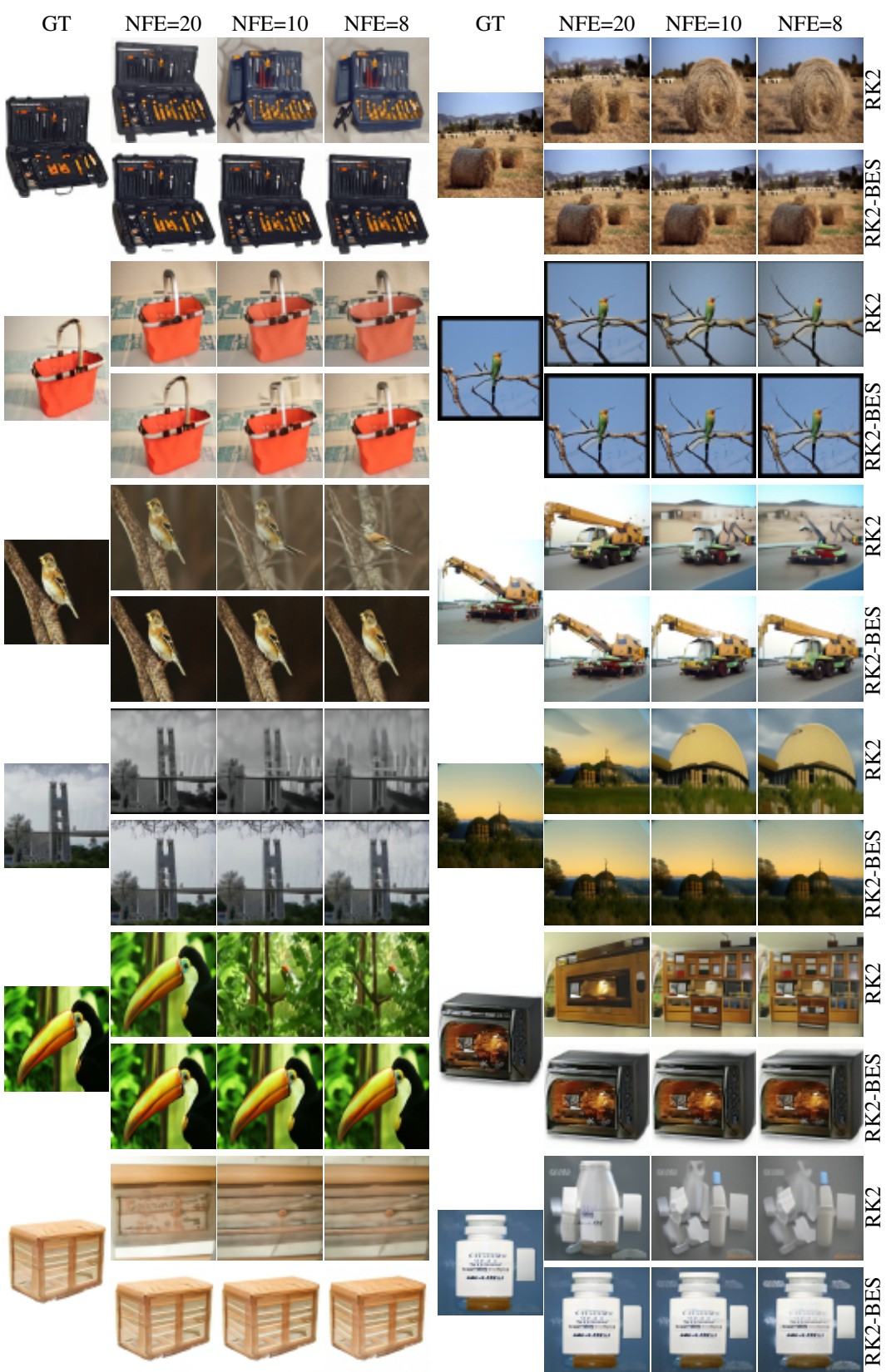

**Figure 28:** Comparison of FM/$v$-CS ImageNet-64 GT samples with RK2 and Bespoke-RK2 solvers.

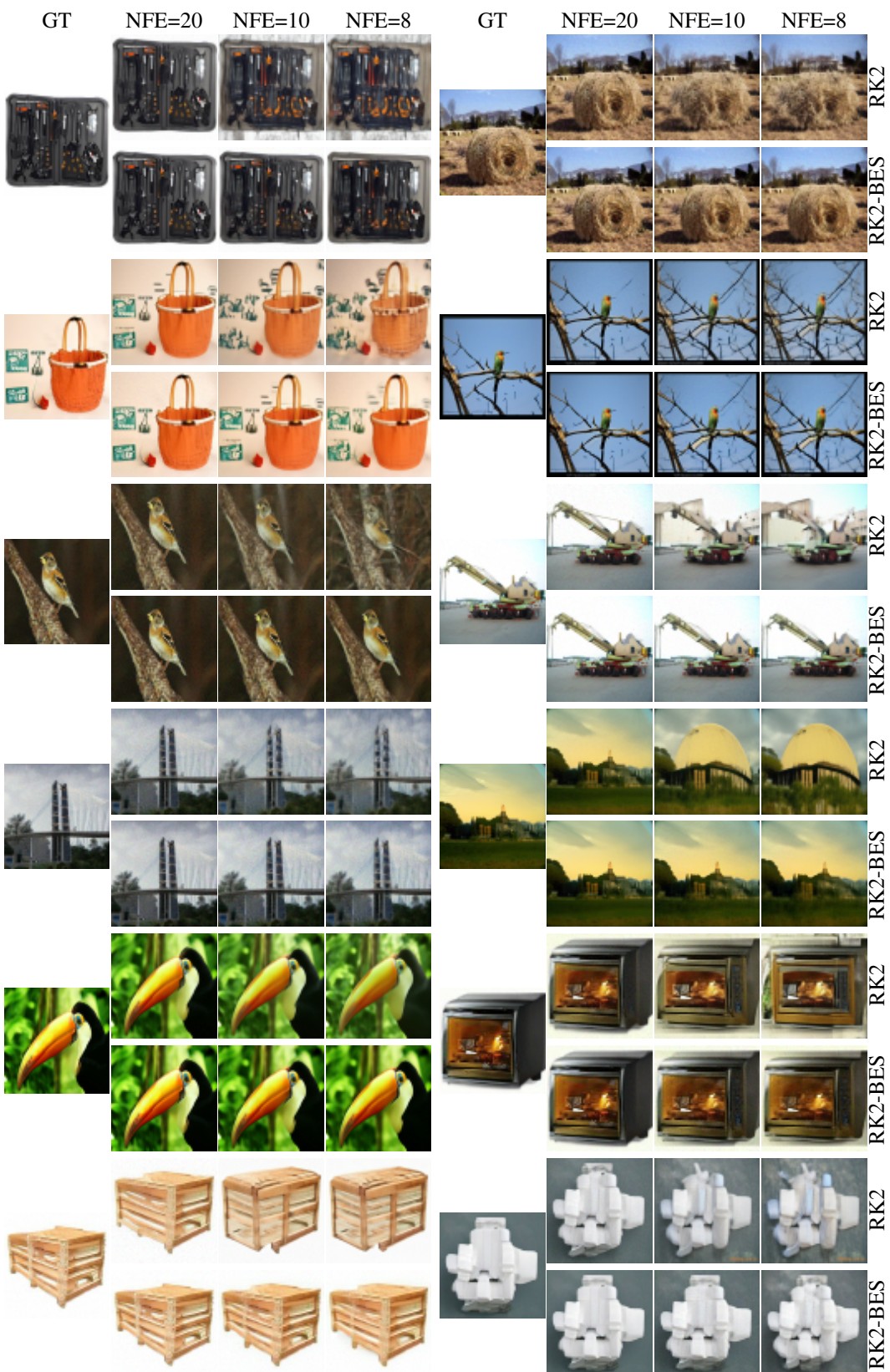

**Figure 29:** Comparison of $\epsilon$-pred ImageNet-64 GT samples with RK2 and Bespoke-RK2 solvers.

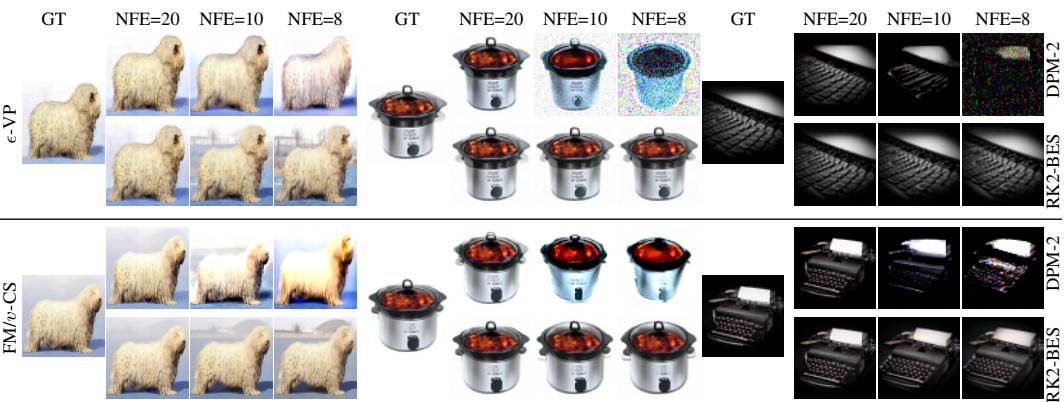

**Figure 30:** Comparison of $\epsilon$-VP and FM/$v$-CS ImageNet-64 samples with DPM-2 and bespoke-RK2 solvers.

