# OpenReview forum: "Bespoke Solvers for Generative Flow Models"
_ICLR.cc/2024/Conference — ICLR 2024 spotlight_

### Official Review · Reviewer_ovsR · 2023-10-29

**Soundness:** 3 good
**Presentation:** 2 fair
**Contribution:** 2 fair
**Rating:** 6
**Confidence:** 3

**Summary:**

The paper proposes a framework for creating custom ODE solvers tailored to the ODE of a given pre-trained flow model. The framework is efficient in terms of additional parameters and training times and can produce high-quality images with a low number of function evaluations.

**Strengths:**

- The method is sound and demonstrates good results in fast sampling.
- The training time is short, and the number of parameters is low.

**Weaknesses:**

**Major concerns**
- Both the quality and the speed of sampling (the number of function evaluations) of the proposal are not as competitive as distillation methods.
- Karras et al., 2022 [1], propose a time discretization method to determine $t_i$, achieving an FID of 1.97 with unconditional image generation on CIFAR10 when NFE = 35. Your proposed method also seeks the sequence of $t_i$, so I strongly compare it with Karras's method.
- Minor Point: There is still room for improvement in the presentation of the paper, especially in notations and function definitions in Section 2.1. It takes me a considerable amount of time to understand.

**Questions:**

- Is proposing a learnable transformed path instead of fixed and cherry-picking ones, as in previous works, the main contribution of the paper?
- Does increasing NFE to 35 boost the performance of your method to be comparable to EDM in [1]? More generally, what happens to your method's performance when NFE is larger than 20? The same questions apply in the case of FID < 10.
- Can your method possibly be applied to solve Stochastic Differential Equations (SDE) directly?

**References**

[1]  Tero Karras, Miika Aittala, Timo Aila, and Samuli Laine. "Elucidating the design space of diffusion-based generative models." Advances in Neural Information Processing Systems, 2022.

---

> ### Author Response · Authors · 2023-11-22
> **Authors' response**
>
> We thank the reviewer for dedicating the time to understand our paper and for raising fundamental questions regarding the paper.
>
> > Both the quality and the speed of sampling (the number of function evaluations) of the proposal are not as competitive as distillation methods.
>
> We respectfully disagree, for the following reasons:
> 1. Regarding quality, we reach within 1%-5% of GT FID sampling quality for each pre-trained model we have used which provides indistinguishable results as can be seen, e.g., in Figures 6 and 7. It is true however we may have not used the best performing pre-trained models out there, but we certainly use competitive models for the datasets we considered.
> 2. Distillation does not guarantee sampling from the original model and in fact **changes the distribution** by updating the model parameters with fine-tuning. In practice, effective distillation requires **access to training data**, where bespoke solvers do not. As far as we are aware, distillation without access to training data, using only model samples, are not as performant.
> 3. Bespoke is trained on a fraction of the compute time (i.e., 1% of the GPU time)  and parameters (80 params in total) and does **not change the pre-trained model’s distribution**. Furthermore Bespoke solvers are guaranteed to samples from original distribution with sufficient NFE (see Theorem 2.2).
> 4. Following the reviewer’s question we **added a comparison to distillation in our framework**, see Appendix K.4. Note that for a fair comparison we use the same amount of data (GT samples from the pre-trained model) and same compute budget to train the distilled model. As seen in Figure 19 the distilled model cannot generalize as well as Bespoke to validation samples and is significantly inferior in terms of quality vs. NFE.
>
> > Is proposing a learnable transformed path instead of fixed and cherry-picking ones, as in previous works, the main contribution of the paper?
>
> It’s a big part of it, but not the only contribution. We would summarize our contributions as follows:
> 1. First example (as far as we know) of a differential family of parametric solvers. The central idea is to define it via learnable path transformations (as the review mentioned).
> 2. A proof that scale-time transformations cover all Gaussian schedulers, justifying our focus on these transformations in the paper.
> 3. Open the door to further improve the solvers by using our framework with more elaborate path transformations $\varphi_r$,$t_r$, e.g., making them conditional or taking $\varphi_r$ change per pixel.
> 4. A tractable step-parallel loss for minimizing the global truncation error (i.e., RMSE loss).
>
>
>
> > Karras et al., 2022 [1], propose a time discretization method to determine t_i, achieving an FID of 1.97 with unconditional image generation on CIFAR10 when NFE = 35. Your proposed method also seeks the sequence of t_i, so I strongly compare it with Karras's method.
>
> We would like to draw the attention of the reviewer to the fact **we already compared to EDM sampling method** in Figure 4. We used the **exact same baseline model published by EDM** (see Figure 2a the EDM paper [1]) and applied our method to this pre-trained model: While EDM achieves FID=3.01 with NFE=35, which is %3 from the model’s best FID (i.e, with NFE=1024), Bespoke solvers achieves FID=2.99 already at NFE=20. We essentially just added our method’s curve to Figure 2a in the EDM paper [1].
>
>
> It’s true that the EDM paper also suggested an improved training procedure for diffusion models and that was used to train an even better CIFAR model, as is shown in Table 2 in EDM paper [1]. However note that only the first row (denoted by “Baseline” in Table 2 in [1]) is a result of EDM’s sampling method, all other rows in that table **required retraining the model**.
>
> [1] Tero Karras, Miika Aittala, Timo Aila, and Samuli Laine. "Elucidating the design space of diffusion-based generative models." Advances in Neural Information Processing Systems, 2022.
>
> > Does increasing NFE to 35 boost the performance of your method to be comparable to EDM in [1]? More generally, what happens to your method's performance when NFE is larger than 20? The same questions apply in the case of FID < 10.
>
> (We assume the reviewer means NFE<10 in their question.) As mentioned in previous answer we already **match EDM performance for their published baseline model** for NFE=20 and therefore no need to use higher NFE. Following the reviewer’s request we **added more NFE (large and small) to our main graphs and tables**, see Figure 5 and Table 2.
>
>
> >  Can your method possibly be applied to solve Stochastic Differential Equations (SDE) directly?
>
> That’s an excellent question. We don’t know yet, but feel this is a good potential direction for future work. It may be worth noting that the majority of fast sampling methods use an ODE formulation.

---

> > ### Comment · Reviewer_ovsR · 2023-11-23
> >
> > Thank you for your response. Your answer and the revision clarify my concerns about the paper. I raise the score from 5 to 6.

---

### Official Review · Reviewer_KhFN · 2023-10-30

**Soundness:** 3 good
**Presentation:** 3 good
**Contribution:** 4 excellent
**Rating:** 8
**Confidence:** 3

**Summary:**

This paper presents a distillation approach diffusion models that is based on ODE solvers. The family of Runge-Kutta ODE integrators is used as a basis for performing a "time integration" for the diffusion time with learned coefficients for the different time steps. The number of time steps is chosen quite low such that correspondingly a low number of function evaluations (NFEs) of the pre-trained diffusion model is needed. A small network is trained that parametrizes the time integrator, with the benefit of a small number of weights and fater training.

The NFE count is used as a central "dimension" along which the RMSE and FIG performance of different versions is evaluated given a certain number of NFEs. The approach seems to achieve a very good performance with relatively low NFE and fast training.

The shown samples are close to the "GT" versions (with larger NFEs from the original model), and seem to perform better than running an RK-integrator directly on the pre-trained model. The images are typically shown for 20,10 and 8 evaluations, and the versions of the proposed method usually show few variations, i.e. stay close to the GT version. This is of course raises the question how even fewer steps would perform, and when the visual difference would start to grow. This is unfortunately not (yet) demonstrated, but should be easy to add.

**Strengths:**

The paper targets an important area, the runtime of generative models from diffusion approaches. The distillation via a parametrized and learned ODE is a new idea as far as I can tell, and the results are convincing. The quantified performance of the models as well as the qualitative examples look very good to me.

Overall, I find the results convincing, and would suggest an "accept" given my current understanding of the work. There are a few smaller open points below which I hope the authors can clarify in the rebuttal.

**Weaknesses:**

While the main approach with custom RK integration makes a sound impression, the loss from section 2.3 seems a bit weaker in comparison. It essentially seems to yield a weighting the the MSE terms with the M_i coefficients, and replacing the acutal Lipschitz constants with 1 everywhere seems ad-hoc. I was also missing an ablation showing how much the training benefits from this loss over a "regular" RMSE loss.

**Questions:**

1) I was surprised about the statement that best FID iterations are "reported", but best RMSE iterations is shown. So the results shown in figures 6,7,... are from a different model than the graphs and tables shown? Can the authors can explain why?

2) The introduction claims a "very small number of learnable" parameters, but I didn't find a table listing the actual parameter counts of the original models and the additional parameters for the "bespoke" versions. Can the authors provide these?

3) Can the authors re-run some cases of "bespoke" training with a regular RMSE loss, e.g., using GT values from a more finely sampled reference trajectory? This would highlight the influence of the loss.

4) Can the authors show some results from models with fewer steps than 8? E.g., 6, 4 or 2 steps?

----

I think the authors have done a nice job addressing the remaining open questions in their rebuttal, and included a collection of additional background information and illustrative results. As such, in line with my previous score, I would recommend an accept for this paper.

---

> ### Author Response · Authors · 2023-11-22
> **Authors' response**
>
> We thank the reviewer for their positive review. Below we address the main questions and concerns from the review.
> > I was surprised about the statement that best FID iterations are "reported", but best RMSE iterations is shown. So the results shown in figures 6,7,... are from a different model than the graphs and tables shown? Can the authors can explain why?
>
> All qualitative and quantitative results are taken from the same pre-trained model and bespoke solver; the difference is in the stopping criteria for the training of the bespoke solver used to report results, as RMSE and FID do not correlate perfectly. For qualitative results (Tables, Graphs) we reported results with the best FID iteration following the common practice in previous works. For sampling we wanted to test our best RMSE iteration. However, following the reviewer's question we also **added in Figure 20 in Section K.5** a comparison of the two stopping criteria (best FID and best RMSE) for the main images in the paper. As can be seen, they are practically indistinguishable.
>
> > The introduction claims a "very small number of learnable" parameters, but I didn't find a table listing the actual parameter counts of the original models and the additional parameters for the "bespoke" versions. Can the authors provide these?
>
> Thanks for highlighting this. We **added the total param count in Table 4 in Appendix F**. In addition, as stated right before equation 21 the number of learnable parameters of Bespoke-RK2 solver is $p=8n-1$, where $n$ is the number of steps. For example, a Bespoke-RK2 solver with 4/5/8/10 steps uses 31/39/63/79 parameters (resp.), whereas in contrast our smallest pre-trained model (CIFAR10) has over 55M parameters.
>
> > Can the authors re-run some cases of "bespoke" training with a regular RMSE loss, e.g., using GT values from a more finely sampled reference trajectory? This would highlight the influence of the loss.
>
> As we are not sure what the reviewer means by “regular RMSE loss” we have **added a loss ablation section (Appendix K.1)** where we compare three versions of the Bespoke loss: the RMSE loss from equation 6, the RMSE-Bound from equation 26, and the loss in equation (26) with just taking $M_i=1$ for all $i$. We call the latter loss the Local Truncation Error (LTE) loss. See Algorithm 5 for all three variations of Bespoke solver learning, and Figure 15 for their results on ImageNet 64.
> The non-parallel RMSE loss requires keeping in memory the full computational graph of Algorithm 2 and consequently we needed to use Activate Checkpointing to reduce memory consumption in order to be able to run this loss. (We **added a discussion in the beginning of Section 2.3** on this issue.) As can be seen in the graphs, RMSE as expected reaches lowest RMSE, the second best is the RMSE-Bound loss and worst in terms of RMSE is the LTE. As for FID, RMSE performs worst, while for FM-OT model RMSE-Bound and LTE perform equivalently for NFE>10, and LTE has advantage as far as FID goes otherwise. Since our goal is to reduce RMSE as much as possible and provide a memory-scalable training algorithm we opted to use the memory efficient RMSE-Bound in the paper.
>
> > Can the authors show some results from models with fewer steps than 8? E.g., 6, 4 or 2 steps?
>
> Yes. Note that $n$ steps correspond to $2n$ NFE since we use RK2, and the paper already has 4 steps (8 NFE) and above. Therefore, following the reviewer’s request we **added 2 steps (4 NFE) and 3 steps (6 NFE), see Figure 5**. Note that while at 4 NFE Bespoke does not provide a good FID it already reduces the RMSE by more than 50% over the runner-up baseline’s RMSE.

---

### Official Review · Reviewer_Zfm7 · 2023-11-04

**Soundness:** 4 excellent
**Presentation:** 3 good
**Contribution:** 4 excellent
**Rating:** 8
**Confidence:** 4

**Summary:**

This paper introduces the Bespoke solvers for generative flow models, which is customized to the pretrained flow models. To apply this, a model that returns a small number of parameters (<80) is introduced to learn the hyperparameters for the consistent ODE solvers. To define the parametric family of solvers, the transformed sampling path from an invertible transformation $\varphi_r$ is first applied.

For transforming the sample paths, two components are applied: the time reparameterization $r\to t_r$ and the invertible transformation $\varphi_r$. This framework finally turns into the generalized noise scheduler for all possible diffusion models and flow models, by normalizing the inputs with the invertible transformation and denoting the timestep with the time parameterization function. This is realized by learning the hyperparameters that consists the scaling function $s(t)$ and time parameterization function $t_r(t)$.

Next, the tractable RMSE loss that computes the global truncation error between the approximate sample and the ground truth (GT) sample is enabled: the upper bound of the RMSE loss is formulated by the weighted sum of the local truncation errors, weighted with the Lipschitz constants. By minimizing this RMSE loss, one can reduce the gap between the approximate sample (given by the solver) and the ground truth (given by the oracle ODE solver).

**Strengths:**

(1) This method solves the important problem in the diffusion/flow models, the time-step and input scaling problem by learning hyperpamareters of the pretrained model ubiquitously, by optimizing the time steps and the input scaling with just some data-driven optimization with the pretrained model. If trained properly, and the bound between RMSE loss and the global truncation loss is rightly narrowed, then sampling from this learning-base parameterization derives good sampling results, as the paper proposes.

(2) The writing is compact and sound; one can easily understand why the learning-based parameterization is required and how this benefits the sampling process of the diffusion/flow models, with abundant experimental results and theoretical supporting materials.

(3) According to Table 3, this method requires much less training time of the hyperparameters (about 0.5~2% of the original training time) compared to the existing distillation-based method for fast sampling.

**Weaknesses:**

(1) Even though the RMSE objective is upper bounded by the Lipschitz loss, there can be some gap between these two loss; unless the loss converges to zero.

(2) There results are not yet validated with larger-scale (than size $64\times 64$) datasets, like FFHQ or LSUN.

**Questions:**

(1) If this bespoke solver is trained well with the higher-order solver, it is curious about the optimal required order of the solver with the learning-based coefficients: this means, if the higher-order solver is not required, the higher-order coefficient $t_r ''$ (if exists) or $s_r ''$ is expected to have low value, which is nearer to zero.

(2) Is there any ablation with the $s_i$-only or $t_i$-only cases? (In this cases, $t_i$ is uniform (or EDM-like) or $s_i$ follows the VP, VE, or EDM preconditioning...) This additional will help understanding the advantages of training these hyperparameters.

**Details Of Ethics Concerns:**

None.

---

> ### Author Response · Authors · 2023-11-22
> **Authors' response**
>
> Thank you for the positive review and interesting questions/comments. We address your questions/comments below.
>
> > Even though the RMSE objective is upper bounded by the Lipschitz loss, there can be some gap between these two loss; unless the loss converges to zero.
>
> True, after overcoming a memory consumption issue with the RMSE loss (using active checkpointing trading memory footprint with computation) we also added a comparison for training with the direct RMSE loss in Appendix K.1. As expected the direct RMSE loss is able to introduce a further improvement in RMSE compared to the RMSE-Bound loss, however it is surprisingly worse in terms of FID and is slightly less stable to train.
>
>
> > There results are not yet validated with larger-scale (than size 64×64) datasets, like FFHQ or LSUN.
>
> This is actually not true: we also validate on 128x128 and 256x256 models: In Figures 5, 7 and Table 2 we report (qualitative and quantitative) results on ImageNet 128x128; and in Figures 7, 14 we report (qualitative and quantitative)  results for AFHQ 256x256.
>
>
> > If this bespoke solver is trained well with the higher-order solver, it is curious about the optimal required order of the solver with the learning-based coefficients: this means, if the higher-order solver is not required, the higher-order coefficient $t_r″$ (if exists) or $s_r″$ is expected to have low value, which is nearer to zero.
>
> We build our parametric family of solvers for training a Bespoke solver by composing a scale-time transformation of the velocity field and a generic base ODE solver (e.g., Euler, Midpoint). In Theorem 2.2 we prove that the order of all members of the resulting parametric family of solvers is the same as the **base ODE solver**. Hence, the order of the resulting parametric family of solvers is independent of the transformation of the velocity field and higher derivatives of the transform functions $s_r,t_r$ are not used. Indeed, note that the scale-time transformation of the velocity field given in equation 16 depends only on the values of $t_r$, $s_r$ and their first derivatives.
>
> > Is there any ablation with the $s_i$-only or $t_i$-only cases? (In this cases, $t_i$ is uniform (or EDM-like) or s_i follows the VP, VE, or EDM preconditioning...) This additional will help understanding the advantages of training these hyperparameters.
>
> Yes. As described in Section 4 (Experiments) we conducted exactly such an ablation that compares the three cases:
> 1. Train both $t_i, \dot{t}_i$ and $s_i, \dot{s}_i$.
> 2. Train only $t_i, \dot{t}_i$ (i.e., keep  $s_i=1, \dot{s_i}=0$ for all $i$).
> 3. Train only $s_i, \dot{s}_i$ (i.e., keep  $t_i=ih, \dot{t}_i=1$ for all $i$).
>
> The results of this ablation are reported in appendix K.2 in figure 16. To answer the reviewer’s question regarding the learned $t_i$ and $s_i$ in these cases we add their learned schemes in figure 17. We don't know the analytic relation of the minimizers of our loss (i.e., learned $s_i$ and $t_i$) to previous ODE parameterizations, but this is an interesting question for future work. Lastly, we compare to EDM in Figure 4, where it is shown our method outperforms EDM sampling method.

---

### Official Review · Reviewer_7aEk · 2023-11-06

**Soundness:** 3 good
**Presentation:** 4 excellent
**Contribution:** 3 good
**Rating:** 8
**Confidence:** 3

**Summary:**

This work introduces Bespoke Solvers, a method for efficiently sampling flow models by optimizing an instance-specific solver for the flow's particle ODE. Given a fixed budget of $n$ function evaluations, fitting a bespoke solver involves optimizing over a parametric family of $n$-point discretization schemes to minimize the average discretization error over new sample generations. To identify the relevant parametric family, the authors derive how a generic ODE solver changes under space- and time-reparametrization of the underlying particle trajectories. While these reparametrizations may be complicated functions of the underlying data, it is sufficient to know their values and derivatives _only_ at the $n$ discretization points, so that optimizing the bespoke solver only requires fitting $O(n)$ parameters.

Minimizing the global discretization error directly may be computationally challenging due to the recursive dependence of the errors on previous time steps. Instead, the authors propose to minimize a weighted sum of one-step discretization errors, which can be computed at different times independently (and hence in parallel) and which is also an upper bound on the discretization error when the weights are chosen appropriately. The proposed method is shown to outperform existing dedicated solvers and it is shown to be competitive with distillation-based methods (ex. Ho and Salimans 2022) while requiring substantially less training time and fewer parameters.

**Strengths:**

- Clarity: the proposed method is elegant and it is well-explained in the paper. The derivations are correct to the best of my knowledge.
- Highly practical and effective: the method is also cheap to train, adaptable to many existing architectures, and it leads to significant benefits for simulating flow-based image samplers. The experiments in Tables 1 and 2 are especially convincing

**Weaknesses:**

- No tunability for NFE: one minor weakness of this approach is that training the bespoke solver requires choosing up-front the number of function evaluations to be used in sampling. In contrast, non-instance dependent schemes can adjust to different NFE budgets at sample time, or they can be run 'until convergence' (choosing NFE adaptively for each particle trajectory).

**Questions:**

- Do you have any understanding of the limiting behavior of the bespoke solver parameters as NFE is increasing? In Figure 18, the parameters seem to be converging to a nontrivial limit as NFE increases. This is related to my comment in 'Weaknesses,' since one way to tune NFE could be to identify the limit of the solver parameters and to discretize it 'on demand' for different choices of NFE.
- The scale-time parametrization seems like an arbitrary choice. Are there any benefits to using more complicated parametrizations? For example, did you run any experiments with time-dependent affine transformations (eg. adding an additive bias parameter to equation 14) or with higher order polynomials?

---

> ### Author Response · Authors · 2023-11-22
> **Authors' response**
>
> We thank the reviewer for their thoughtful review. Below we address the comments and questions raised in the review.
>
> > No tunability for NFE: one minor weakness of this approach is that training the bespoke solver requires choosing up-front the number of function evaluations to be used in sampling. In contrast, non-instance dependent schemes can adjust to different NFE budgets at sample time, or they can be run 'until convergence' (choosing NFE adaptively for each particle trajectory).
>
> Thanks, this is indeed a good point and interesting future work. For now, we add it as a limitation in Section 5.
>
>
> > Do you have any understanding of the limiting behavior of the bespoke solver parameters as NFE is increasing? In Figure 18, the parameters seem to be converging to a nontrivial limit as NFE increases. This is related to my comment in 'Weaknesses,' since one way to tune NFE could be to identify the limit of the solver parameters and to discretize it 'on demand' for different choices of NFE.
>
> Again, good point, thanks. We currently don’t have an understanding of the limit behavior of the parameters $s_r,t_r$ as NFE increases. Both finding limit behavior and/or optimization of universal (i.e., arbitrary NFE)  rules are worthy future directions.
>
>
> > The scale-time parametrization seems like an arbitrary choice. Are there any benefits to using more complicated parametrizations? For example, did you run any experiments with time-dependent affine transformations (eg. adding an additive bias parameter to equation 14) or with higher order polynomials?
>
> One motivation/justification for the scale-time transformation is given in theorem 2.3: It shows scale-time transformations cover all possible noise-to-data schedulers. In that sense we optimize over a set of solvers that contains all previous ad hoc solvers defined by particular choices of schedulers. More complicated transformations are very interesting future work. For example: $\varphi_r$,$ t_r$ can be made to depend on condition/guidance weight, and $\varphi_r$ can be a more complicated space transform, e.g., per-pixel transformation. We only started exploring this vast design space. We definitely agree with the reviewer that there are many interesting future work directions.

---

### Official Review · Reviewer_vykk · 2023-11-07

**Soundness:** 3 good
**Presentation:** 3 good
**Contribution:** 3 good
**Rating:** 6
**Confidence:** 4

**Summary:**

This work addresses the issue of slow sampling in Diffusion models. The authors consider deterministic sampling of Diffusion models via the Probability Flow ODE. They address the issue by introducing a learnable (continuous) reparameterization of the ODE. Rather than learning the continuous parameterization directly, they discretize the reparameterized ODE using Runge--Kutta methods, which allows for learning only a discrete set of parameters (e.g. $4n-1$ parameters for Euler's method, where $n$ is the number of steps in the discretization). The parameters can then be learned by minimizing the discrepancy of the learned Runge--Kutta discretization to ground truth trajectories of the Diffusion model (which are in practice simulated using a higher-order adaptive ODE solver). The authors claim that the training time of the reparameterized ODE is roughly 1% of the training time of the original Diffusion model. The authors compare their method to the literature on Cifar-10 and ImageNet (64x64).

**Strengths:**

- The problem of accelerating inference in Diffusion models is important
- The structure of the paper is good, and the paper is generally well written (I am very happy that the authors opted to include Figure 2, and Algorithm boxes 2&3 in the main paper which help to understand the method)
- The idea of learning parameters for a reparameterization of a Neural ODE is original (to the best of my knowledge)
- The authors cover a lot of previous work (however I think some works are misrepresented, see also below)

**Weaknesses:**

**Missing details**: I think some details of the training are missing. How many GT trajectories are used and is the time for computing GT trajectories accounted for when claiming that the method only needs "roughly 1% of the GPU time" compared to training of the Diffusion model? This seems like a very important detail.

**Single guidance value training**: As far as I understand, the authors need to retrain fresh parameters for each guidance value (and compute GT trajectories for each guidance value). I think the authors should have addressed this as drawback of the method (compared to other methods).

**Misrepresentation of Distillation approaches**: I think the authors are misrepresenting the work on Diffusion distillation. For example, they say "Distillation does not guarantee sampling from the pre-trained model's distribution" - while this is correct the same applies to the proposed method. In fact, the inference distribution of a Diffusion model is always coupled to a numerical discretization; any solver will result in a different distribution of the generated samples.

In general, I also think the authors should have tried to compare the performance of their approach to distillation. For example, they could have done Progressive / Consistency distillation using a fixed compute budget. Unfortunately, there are no comparisons at all.

**Interpretation of Figures 17/18/19**: It is actually quite nice that the authors can visualize the learned parameters (since there are so few) but unfortunately the work is missing a discussion on the figures. I think it's quite interesting that $s_r \approx 1$ for most experiments and that $t_r$ seems to increase linearly with $r$. Could the method for example be used to explain the success of the DDIM ODE reparameterization?

**Questions:**

**Depth of Experiments**: Since running the method is quite cheap, it would have been nice to include larger values of $n$ (currently only up to $n=10$) and higher order methods (e.g. RK3). Is there any particular reason why this has not been done?

**Limited Outlook**: I think the idea of repameterizing the ODE with learnable parameters is neat. How could this method potentially be scaled/improved? Are more elaborate reparameterizations possible?

**Guidance scale(s)**: What guidance scale(s) is/are used in the experiments?

Also see some questions entangled with the Weaknesses above.

---

> ### Author Response · Authors · 2023-11-22
> **Authors' response [1/2]**
>
> We thank the reviewer for their insightful comments and questions. Below we address them in two consecutive posts.
>
> > Missing details: I think some details of the training are missing. How many GT trajectories are used and is the time for computing GT trajectories accounted for when claiming that the method only needs "roughly 1% of the GPU time" compared to training of the Diffusion model?
>
> Thank you for your question. We added the number of GT trajectories used for training in Table 3 (Appendix F):
> The Bespoke solvers for CIFAR10, ImageNet 64, ImageNet 128, and AFHQ were trained on 72k, 48k, 48k, 4k trajectories (resp.). And yes, the reported training time does include the time it took to generate the trajectories. In fact, separating GT generation and training can further reduce total time requirement.
>
>
> > Single guidance value training: As far as I understand, the authors need to retrain fresh parameters for each guidance value (and compute GT trajectories for each guidance value). I think the authors should have addressed this as drawback of the method (compared to other methods).
>
> Thanks for the comment. Just to clarify, we simply tune guidance weight for performance prior to training a Bespoke solver, so there is no need to tune guidance weight simultaneously with the Bespoke solver. Nevertheless, for the use case of sampling with different guidance weights we agree this is a limitation and we added this point as a limitation, see Section 5 (Limitations and Conclusions). We further note that one possible solution for changing guidance weights during sampling is to let $t_i$ and $s_i$ depend on the guidance weight, but we left these extensions to future work.
>
>
>
> >  I think the authors are misrepresenting the work on Diffusion distillation. For example, they say "Distillation does not guarantee sampling from the pre-trained model's distribution" - while this is correct the same applies to the proposed method. In fact, the inference distribution of a Diffusion model is always coupled to a numerical discretization; any solver will result in a different distribution of the generated samples.
>
> We don’t fully agree. There is a GT distribution defined by the pretrained model $u_t$ and the ODE in Equation (1). While it’s true that numerical errors in solvers change the distribution of the samples, Bespoke solvers are still **guaranteed** to sample closer and closer to the GT distribution as the number of samples increases (see “consistency of solvers”, Theorem 2.2). This is due to the fact that **Bespoke solvers do not change the pre-trained model’s distribution**, they solely optimize among fixed order ODE solvers and do not update the original pre-trained model’s parameters. For distillation the GT distribution can change without any guarantee. That is, **distillation methods change the pre-trained model**.
>
>
>
> > In general, I also think the authors should have tried to compare the performance of their approach to distillation. For example, they could have done Progressive / Consistency distillation using a fixed compute budget. Unfortunately, there are no comparisons at all.
>
> We added such a distillation experiment with a fixed compute budget in our framework where we let $u_t$ weights change using a Local Truncation Error loss, see Appendix K.4 and Figure 19. We can offer two potential explanations why distillation is not as performant in our experiments: First, the amount of trajectories/iterations we use to optimize Bespoke is not sufficient for effective distillation (it is an extremely small percentage compared to the full training set; see our response to the first question). Second, to our knowledge, successful distillation methods require **access to data**. Training only on generated data distillation is often less effective.
>
>
>
> > Interpretation of Figures 17/18/19: It is actually quite nice that the authors can visualize the learned parameters (since there are so few) but unfortunately the work is missing a discussion on the figures. I think it's quite interesting that $s_r\approx1$ for most experiments and that $t_r$ seems to increase linearly with $r$.  Could the method for example be used to explain the success of the DDIM ODE reparameterization?
>
> We added a discussion of the learned solvers’ parameters in Appendix L  in the revised paper. We also added the learned parameters of the scale-time ablation experiment in Figure 17. Regarding the relation to known solvers such as DDIM, this is a very interesting question which we defer to future work.

---

> > ### Author Response · Authors · 2023-11-22
> > **Authors' response [2/2]**
> >
> > > Depth of Experiments: Since running the method is quite cheap, it would have been nice to include larger values of n (currently only up to n=10) and higher order methods (e.g. RK3). Is there any particular reason why this has not been done?
> >
> > To address the reviewer’s question we added NFE=24 (and also NFE=6,8 following another reviewer’s question) in the ImageNet experiments, please see Figure 5 and Table 2. EDM suggested a convention of reaching within 3% percent of the model’s best (GT) FID as these produce visually equivalent results. Adopting this convention makes further higher values of NFE redundant as the reported values already reach this 3% range. Regarding higher order solvers, there is no limitation there, we focused on 2nd order solver as this seems to be the de-facto popular order choice for solvers (together with 1st order, which is less performant usually as we noticed).  We do expect future work to experiment with Bespoke using other base solvers, including higher order.
> >
> >
> > > Limited Outlook: I think the idea of repameterizing the ODE with learnable parameters is neat. How could this method potentially be scaled/improved? Are more elaborate reparameterizations possible?
> >
> > As described in the conclusion section (Section 5), the most natural way to improve the ODE reparametrization is to make the transforms $\varphi_r$ and $t_r$ more expressive/elaborate. There are several immediate options: (i) adding dependency on the condition variable and/or the CFG weight. (ii) $\varphi_r$ can be chosen as a more complicated space transform, e.g., per pixel.  Ideally, a sufficiently expressive $\varphi_r$ can straighten all the trajectories and make them constant speed in the transformed space (Optimal Transport), allowing perfect 1 NFE sampling with RK1.  Therefore, taking more expressive $\varphi_r,t_r$ could potentially lead to even smaller NFE sampling with solver’s advantages, i.e., fast and low-compute training, no need for training data access, good generalization to unseen paths, and consistency (i.e., guaranteed sampling from GT distribution for sufficiently large NFEs).
> >
> >
> > > Guidance scale(s): What guidance scale(s) is/are used in the experiments?
> >
> > We added the guidance scales used in the experiments in Table 5 in Appendix M. For the reviewer’s convenience we provide them below. We note that all guidance weights were tuned prior to training Bespoke solvers, i.e., we simply trained Bespoke solver on the best guidance weight for each model.
> >
> > | Dataset      | Model     | Guidance Weight |
> > |--------------|-----------|-----------------|
> > | ImageNet 64  | $\epsilon$-VP | $w=0.20$        |
> > | ImageNet 64  | FM$v$-CS  | $w=0.20$        |
> > | ImageNet 64  | FM-OT     | $w=0.15$        |
> > | ImageNet 128 | FM-OT     | $w=0.50$        |
> > | AFHQ 256     | FM-OT     | $w=0.10$        |

---

### Author Response · Authors · 2023-11-22
**General response**

We thank the reviewers for their detailed and thoughtful reviews. In the revised paper, all significant changes are marked by blue. Here we will detail the main changes done to the paper in response to the reviewers’ comments and suggestions. We also addressed the specific questions/comments of each reviewer separately.

Main changes to main paper/appendices and new experiments:
1. We revised section 2.3 to include a clarification regarding our loss. We extended our results in Figure 5 and Table 2 to include more NFE. We revised section 5 (Limitations and Conclusions) to highlight limitations brought up by the reviewers.
2. **Section F, implementation details:** We add a number of missing details regarding our implementation details.
3. **Section K.1, loss ablation:** We added an ablation experiment and discussion of three variations of the Bespoke losses: direct RMSE, RMSE-Bound (used in paper), and Local Truncation Error (LTE).
4. **Section K.2, scale-time transformation ablation:** We expanded our discussion and included visualization of the trained transformation’s parameters when training time-only and scale-only transformations.
5. **Section K.4, distillation experiment:** We added a comparison and discussion to a distillation-type approach in our framework.
6. **Section K.5, training stopping criteria:** We qualitatively compare the two stopping criteria: FID best and RMSE best.
7. **Section L: trained bespoke solvers:** We added a discussion regarding our trained scale-time transformation.
8. **Section M: pre-trained models:** We added a number of missing details regarding our pre-trained models.

---

### Meta-Review · Area_Chair_dh6Q · 2023-12-08

**Metareview:**

This paper aims at accelerating sampling from diffusion models by learning the invertible transformation of data. The reviewers unanimously rated the paper above the acceptance threshold and acknowledged the quality and importance of the submission. The are minor concerns that ODE-based techniques cannot still outperform distillation approaches in terms of NFEs. Nevertheless, the paper is of interest and the AC is happy to recommend accept.

**Justification For Why Not Higher Score:**

The ODE based approaches are often behind distillation based approaches in terms of the number of function evaluations.

**Justification For Why Not Lower Score:**

The significance of the work is of interest to the community.

---

### Decision · Program_Chairs · 2024-01-16

Accept (spotlight)